# Localised dynactin protects growing microtubules to deliver *oskar* mRNA to the posterior cortex of the *Drosophila* oocyte

Ross Nieuwburg[1†], Dmitry Nashchekin[1†], Maximilian Jakobs[2], Andrew P Carter[3], Philipp Khuc Trong[4], Raymond E Goldstein[4], Daniel St Johnston[1*]

[1]The Gurdon Institute and the Department of Genetics, University of Cambridge, Cambridge, United Kingdom; [2]The Department of Physiology, Development and Neuroscience, University of Cambridge, Cambridge, United Kingdom; [3]Division of Structural Studies, Medical Research Council, Laboratory of Molecular Biology, Cambridge, United Kingdom; [4]Department of Applied Mathematics and Theoretical Physics, University of Cambridge, Centre for Mathematical Sciences, Cambridge, United Kingdom

**Abstract** The localisation of *oskar* mRNA to the posterior of the *Drosophila* oocyte defines where the abdomen and germ cells form in the embryo. Kinesin 1 transports *oskar* mRNA to the oocyte posterior along a polarised microtubule cytoskeleton that grows from non-centrosomal microtubule organising centres (ncMTOCs) along the anterior/lateral cortex. Here, we show that the formation of this polarised microtubule network also requires the posterior regulation of microtubule growth. A missense mutation in the dynactin Arp1 subunit causes most *oskar* mRNA to localise in the posterior cytoplasm rather than cortically. *oskar* mRNA transport and anchoring are normal in this mutant, but the microtubules fail to reach the posterior pole. Thus, dynactin acts as an anti-catastrophe factor that extends microtubule growth posteriorly. Kinesin 1 transports dynactin to the oocyte posterior, creating a positive feedback loop that increases the length and persistence of the posterior microtubules that deliver *oskar* mRNA to the cortex.
DOI: https://doi.org/10.7554/eLife.27237.001

**\*For correspondence:**
d.stjohnston@gurdon.cam.ac.uk

[†]These authors contributed equally to this work

## Introduction

Although most tissue culture cells organise a radial array of microtubules from their centrosomes, most differentiated cell-types lose or inactivate their centrosomes, but still create polarised microtubule arrays that play important roles in the establishment of cell polarity, intracellular trafficking and organising the internal architecture of the cell (*Bartolini and Gundersen, 2006*). For example, both *Drosophila* and vertebrate neurons polarise normally without functional centrosomes, and the latter can even regenerate their axons after centrosome ablation (*Stiess et al., 2010*; *Nguyen et al., 2011*). Thus, specialised cells, such as neurons, must use other mechanisms to nucleate microtubules and to organise them into the polarised microtubule arrays that underlie cell function.

Non-centrosomal microtubules play a particularly important role in the *Drosophila* oocyte, where they form a polarised network at stage 9 of oogenesis that directs the localisation of the maternal determinants that define the anterior-posterior axis of the embryo (*Bastock and St Johnston, 2008*). The organisation of the oocyte microtubules ultimately depends on an unknown signal from the follicle cells that induces the formation of complementary cortical polarity domains: an anterior/lateral domain that is defined by the localisation of the Bazooka/Par-6/aPKC complex and a

**eLife digest** Many cells are asymmetric or polarized, which allows them to perform the tasks necessary for an organism to live and grow. In these polarized cells, the top and bottom, left and right, and front and back parts are different from one another. To achieve this, cells actively move molecules to the locations in the cell where they are needed. One type of molecule that is often confined is messenger RNA, or mRNA for short, which carries a portion of the DNA code to other parts of the cell where it can be translated to make a specific protein.

These mRNA molecules are transported by motor proteins, which run along tracks called microtubules that form a network throughout the cell. Each microtubule has a stable 'minus' end and a dynamic 'plus' end, which constantly grows or shrinks. Motor proteins can generally only transport their cargo in one specified direction. For example, the protein Kinesin-1 moves towards the plus end of the microtubule.

In the fruit fly egg, many molecules are asymmetrically arranged, which later dictate how the larvae will develop. For example, a gene called *oskar* is necessary for the development of the back region of the fly embryo, and its mRNA is transported by Kinesin-1 along microtubules towards the plus ends, at the back end of the egg cell. However, it was unclear how the plus ends accumulate at the back of the egg cell in the first place.

Now, Nieuwburg, Nashchekin et al. used live microscopy to watch how growing microtubules and *oskar* mRNA move in fruit fly egg cells. Comparing normal and mutant fruit flies revealed that a large protein complex called Dynactin stabilizes the microtubule plus ends at the back of the cell. This gives Kinesin-1 enough time to carry *oskar* mRNA along the length of a microtubule to its plus end. When one subunit of Dynactin was mutated, the microtubule plus ends became less stable and did not reach all the way to the back of the developing egg. With this mutation, *oskar* mRNA was still transported by Kinesin-1 but delivered to the wrong place.

All together, these experiments provide evidence that the plus ends of microtubules must be controlled so that motor proteins can deliver their cargoes to the correct destination. Future work will determine exactly how Dynactin stabilizes microtubules and whether this is a general mechanism that can also set up polarized microtubule tracks in asymmetric cells, such as nerve cells.
DOI: https://doi.org/10.7554/eLife.27237.002

posterior domain that is marked by Par-1 (*González-Reyes et al., 1995*; *Roth et al., 1995*; *Shulman et al., 2000*; *Tomancak et al., 2000*; *Doerflinger et al., 2006*). Par-1 then acts to exclude noncentrosomal microtubule organising centres from the posterior, so that the majority of microtubules grow with their minus ends anchored to the anterior/lateral cortex (*Doerflinger et al., 2010*; *Nashchekin et al., 2016*). This results in the formation of an anterior-posterior gradient of microtubules in the oocyte, with a weak orientation bias of 60% of the microtubules growing towards the posterior and 40% towards the anterior (*Parton et al., 2011*).

One of the key functions of the oocyte microtubule cytoskeleton is to direct the transport of *oskar* mRNA to the posterior of the oocyte (*Figure 1A*), where it defines the site of assembly of the pole plasm, which contains the abdominal and germline determinants (*Ephrussi et al., 1991*; *Kim-Ha et al., 1991*; *Ephrussi and Lehmann, 1992*). *oskar* mRNA is transcribed in the nurse cells and is then transported along microtubules into the oocyte by the minus-end directed motor, dynein (*Clark et al., 2007*; *Sanghavi et al., 2013*). Once it enters the oocyte, *oskar* mRNA switches motors and is transported by the plus end directed motor protein, kinesin 1, to the posterior pole, where it is translated and anchored to the cortex by Oskar protein (*Markussen et al., 1995*; *Rongo et al., 1995*; *Brendza et al., 2000*; *Vanzo and Ephrussi, 2002*). Tracking of *oskar* mRNA particles in the oocyte reveals that they move in all directions with a similar net posterior bias to the growing plus ends of microtubules (*Zimyanin et al., 2008*; *Parton et al., 2011*). This suggests that *oskar* mRNA is a passive cargo of kinesin 1 and that its destination is determined solely by the arrangement of the microtubule cytoskeleton. In support of this view, computer simulations show that simply allowing microtubules to extend in random directions from ncMTOCs along the anterior and lateral cortex, but not the posterior, is sufficient to give the observed microtubule distribution in the oocyte and to account for the posterior localisation of *oskar* mRNA by kinesin 1 (*Khuc Trong et al., 2015*).

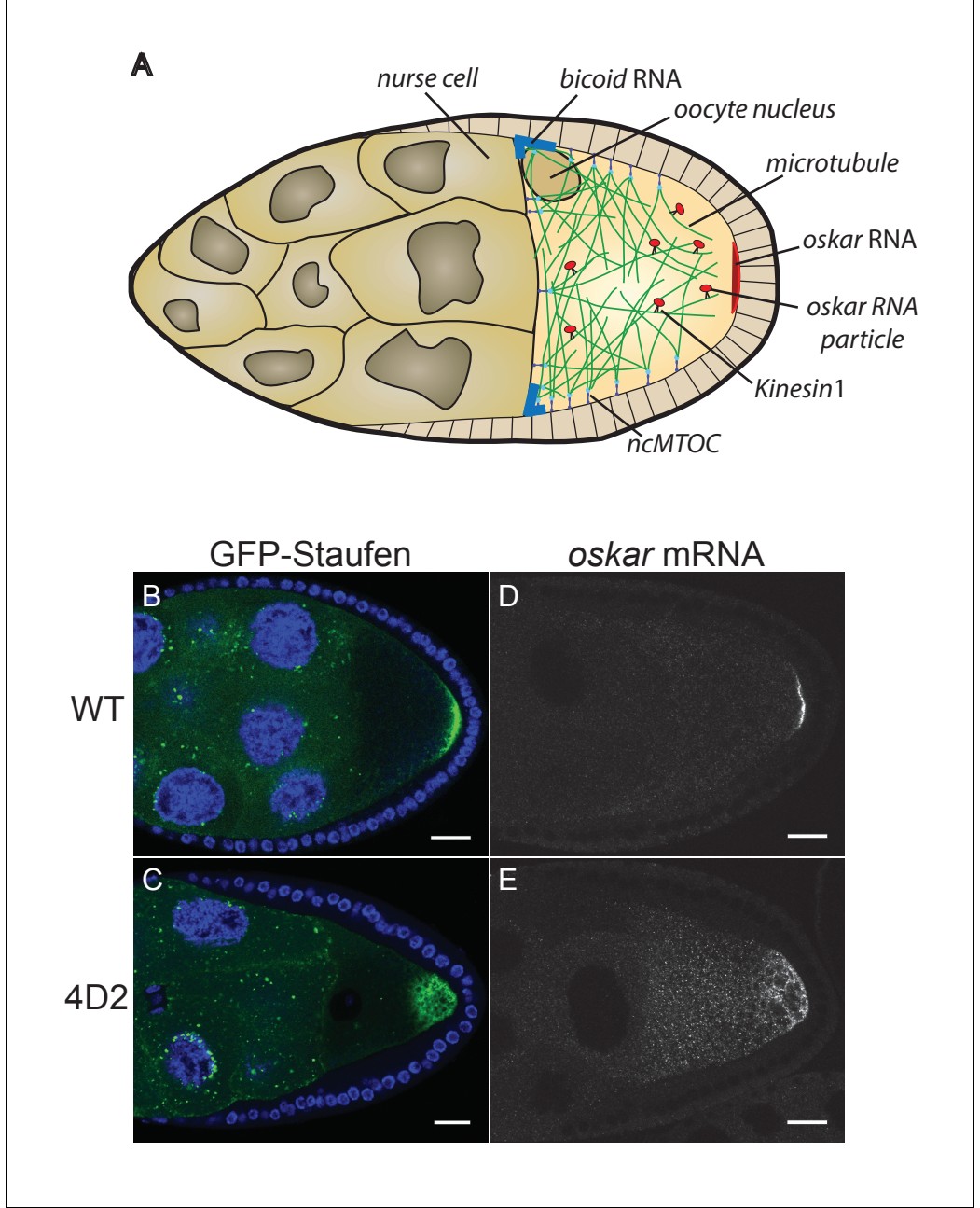

**Figure 1.** The 4D2 mutation disrupts the localisation of *oskar* mRNA at the oocyte posterior. (**A**) A schematic diagram of a stage 9 *Drosophila* egg chamber showing *oskar* mRNA transport. (**B–C**) Confocal images of stage 9 egg chambers from a wild-type ovary (**B**) and from a 4D2 homozygous germline clone (**C**) expressing GFP-Staufen (green) and counterstained with DAPI (blue) to label the nuclei. 31/31 4D2 mutant oocytes showed the same phenotype. (**D–E**) Confocal images of fluorescent in situ hybridisations (FISH) to endogenous *oskar* mRNA in a wild-type egg chamber (**D**) and an egg chamber with a 4D2 homozygous germline clone (**E**). 10/10 4D2 mutant oocytes showed the same phenotype. Scale bar 20 µm.

DOI: https://doi.org/10.7554/eLife.27237.003

The computer simulations treat microtubules as static rods, and do not take into account the dynamic nature of the microtubules in the oocyte, in which the growing plus ends undergo catastrophes when they transition into rapid shrinking, especially on hitting solid surfaces, such as the cortex (*Zhao et al., 2012*). Here, we show that just controlling the distribution of microtubule minus ends is not enough to explain the delivery of *oskar* mRNA to the posterior cortex when the microtubules

are dynamic, and describe an additional layer of control, in which dynactin regulates the microtubule plus ends to increase their persistence specifically at the posterior of the oocyte.

## Results

We have previously identified factors that regulate *oskar* mRNA localisation by performing large genetic screens in germline clones for mutants that disrupt the posterior localisation of GFP-Staufen, a dsRNA-binding protein that associates with *oskar* RNA throughout oogenesis (*St Johnston et al., 1991*; *Ramos et al., 2000*; *Martin et al., 2003*). One mutant from this screen (4D2) gave an unusual phenotype, in which both Staufen and *oskar* mRNA accumulate in the posterior cytoplasm rather than forming a tight crescent at the posterior cortex (*Figure 1B–E*). This phenotype could reflect a failure to anchor *oskar* mRNA to the cortex or a defect in *oskar* mRNA translation, since Oskar protein anchors its own RNA (*Vanzo and Ephrussi, 2002*). Staining for Oskar protein revealed that the small amount of mRNA that is at the posterior cortex is translated, whereas the diffusely-localised cytoplasmic RNA is not (*Figure 2A and B*). Furthermore, the Staufen and Oskar protein that contact posterior cortex remain anchored there at later stages, whereas the higher concentration of protein in the posterior cytoplasm disappears, presumably because it is washed away by cytoplasmic flows (*Figure 2C and D*). Thus, the phenotype does not appear to be a consequence of an anchoring or translation defect, suggesting that the 4D2 mutation disrupts the delivery of *oskar* mRNPs to the posterior cortex.

To test whether the 4D2 phenotype is specific to *oskar* mRNA, we examined whether other molecules that are transported to the posterior by Kinesin 1 are also affected. We first tested a fusion between the motor domain of the kinesin heavy chain and β-galactosidase (Kin-βgal) that behaves as a constitutively active motor (*Clark et al., 1994*). In wild-type egg chambers, Kin-βgal forms a crescent at the posterior of the oocyte at stage 9 (*Figure 3A*). In 4D2 germline clones, however, Kin-βgal localises to a diffuse cloud near the posterior cortex, showing a very similar defect to *oskar* mRNA (*Figure 3B*). Kinesin 1 also transports the dynein/dynactin complex to the posterior, although the function of this localisation is not known (*Li et al., 1994*; *Palacios and St Johnston, 2002*). In 4D2 germline clones, dynein and the dynactin component, p150[Glued], show an identical posterior localisation phenotype to *oskar* mRNA (*Figure 3C–E*). Thus, 4D2 causes a general defect in kinesin-dependent transport to the posterior cortex of the oocyte. As Kin-βgal and the dynein/dynactin complex are not anchored at the posterior, these observations also rule out the possibility that the phenotype arises from a lack of anchoring.

Recombination mapping and fine scale deletion mapping showed that the 4D2 chromosome carries a single lethal mutation within the 66 kb region that is removed by Df(3R)Exel6166, but not by Df(3R)ED5612 (*Figure 3—figure supplement 1*). Crosses to the lethal mutations in this interval revealed that 4D2 fails to complement the *arp1*[c04425] and *arp1*[G3709] alleles of the *Drosophila* dynactin component, Arp1, indicating that 4D2 is a new *arp1* allele. Consistent with this, *arp1*[4D2] contains a missense mutation that changes a highly conserved glutamate residue (E53) to lysine (*Figure 4A*). We also crossed *arp1*[4D2] to two hypomorphic, EMS-induced alleles, *arp1*[1] and *arp1*[2] (*Haghnia et al., 2007*). Some of the transheterozygotes were viable and showed a similar diffuse posterior localisation of GFP-Staufen to *arp1*[4D2] homozygous mutant germline clones, confirming that the mutation in Arp1 is the cause of the phenotype (*Figure 3—figure supplement 2*).

The dynactin complex acts as a cargo adaptor complex for dynein (*Holleran et al., 2001*; *Muresan et al., 2001*; *Zhang et al., 2011*), enhances the processivity of dynein movement along microtubules (*McKenney et al., 2014*; *Schlager et al., 2014*) and also binds to microtubule plus ends through p150[Glued] (*Akhmanova and Steinmetz, 2015*; *Duellberg et al., 2014*; *Lazarus et al., 2013*). Arp1 polymerises into a filament composed of eight copies of Arp1 and one β-actin subunit that forms the backbone of the dynactin complex (*Urnavicius et al., 2015*; *Figure 4B*). The E53 to K mutation in *arp1*[4D2] falls in the loop of subdomain two that mediates the interaction between Arp1 subunits. It is not predicted to affect protofilament formation, because it points away from the interaction interface (*Figure 4C*). However, the E53K mutation should disrupt the interaction of Arp1E subunit with one of the extended regions of p50 dynamitin that anchor the p150[Glued] shoulder to the Arp1 rod (*Figure 4D and E*). This suggests that the mutation does not disrupt the whole dynactin complex, but may alter the conformation or activity of the p150[Glued]/dynamitin/p24 shoulder domain.

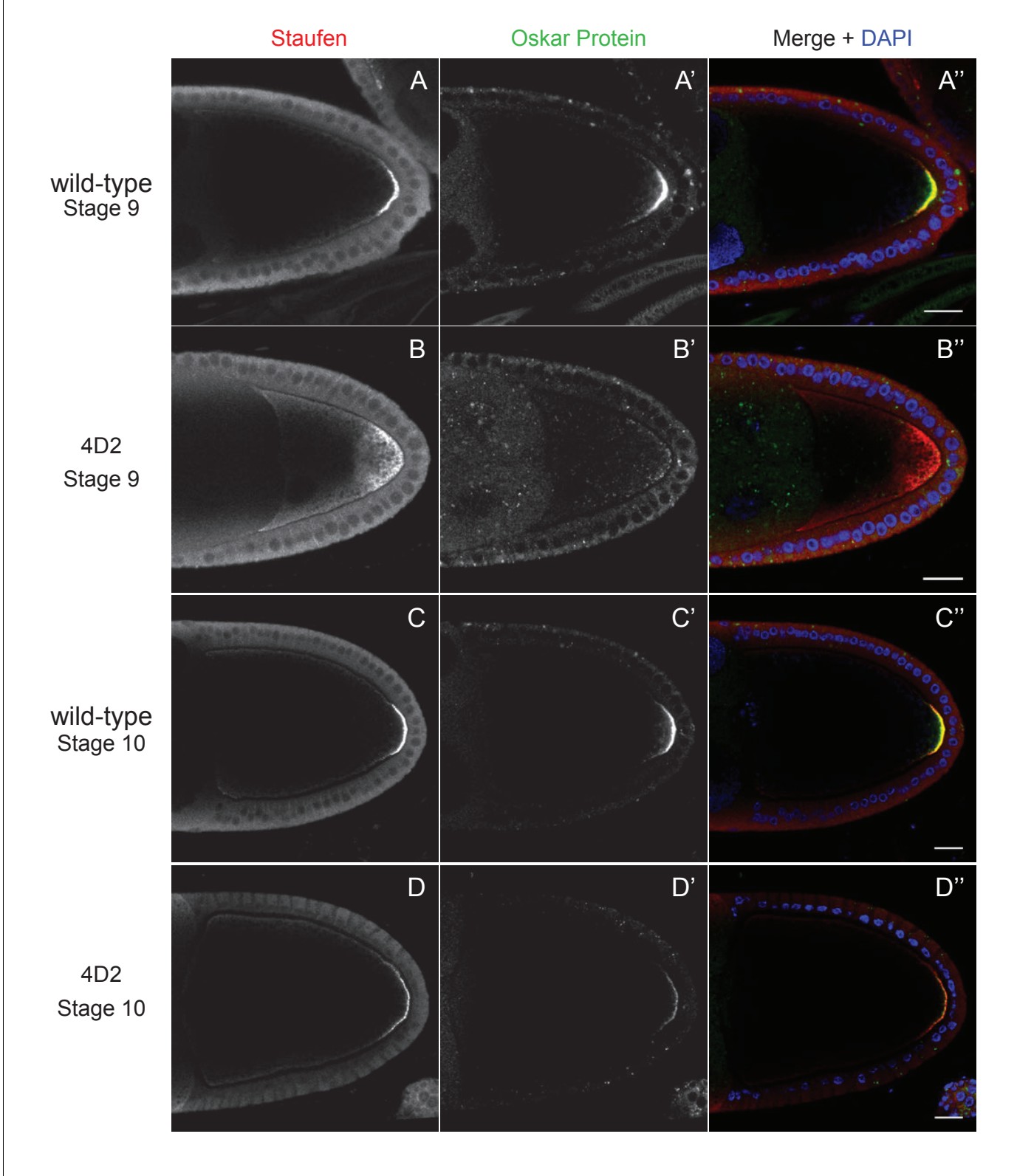

**Figure 2.** The 4D2 mutation does not affect the anchoring of Staufen and *oskar* at the posterior cortex. (A–D) Confocal images of stage 9 (A and B) and stage 10b egg chambers (C and D) from wild-type (A and C) and 4D2 germline clone ovaries (B and D) stained for Staufen (A–D) and Oskar (A'–D') proteins. The merged images (A"–D") show Staufen in red, Oskar in green and DNA in blue (DAPI staining). Although Staufen/*oskar* mRNA complexes are diffusely localised in the posterior cytoplasm, Oskar protein is only made at the posterior cortex. Staufen and Oskar proteins remain anchored at

*Figure 2 continued on next page*

*Figure 2 continued*

the posterior cortex later in oogenesis. 18/18 stage 9 and 21/21 stage 10 4D2 mutant egg chambers showed the same Staufen and Oskar localisation phenotype. Scale bar 20 μm.

DOI: https://doi.org/10.7554/eLife.27237.004

Early in oogenesis, dynein and dynactin are required for the transport of oocyte determinants into one cell of the 16-cell germline cyst, and null mutants in components of either complex therefore fail to form an oocyte (*McGrail and Hays, 1997*; *Liu et al., 1999*; *Bolívar et al., 2001*; *Mirouse et al., 2006*; *Haghnia et al., 2007*). The oocyte is specified normally in *arp1*[4D2] germline clones, however, as shown by the accumulation of Orb in one cell of the 16-cell germline cyst in region 3 of the germarium (*Figure 5A and B*). By contrast, the null mutation, *arp1*[c04425], completely blocks oocyte determination to give rise to cysts with 16 nurse cells (*Figure 5C*). This indicates that *arp1*[4D2] does not disrupt the function of the dynactin complex as an activator and cargo adaptor for dynein.

Dynein and dynactin both fulfil multiple functions during the later stages of oogenesis: they are required to transport RNAs, such as *bicoid*, *gurken* and *oskar* from the nurse cells into the oocyte (*Clark et al., 2007*; *Mische et al., 2007*), they transport *bicoid* mRNA to the anterior of the oocyte (*Duncan and Warrior, 2002*; *Weil et al., 2006*; *Weil et al., 2008*; *Trovisco et al., 2016*), anchor the nucleus at the dorsal/anterior corner of the oocyte (*Swan et al., 1999*; *Lei and Warrior, 2000*; *Januschke et al., 2002*; *Zhao et al., 2012*), and localise and anchor *gurken* mRNA above the nucleus (*MacDougall et al., 2003*; *Delanoue et al., 2007*). Both *bicoid* and *oskar* mRNAs enter the oocyte normally in *arp1*[4D2] homozygous germline clones, and *bicoid* mRNA localises to the anterior cortex as in wild-type (*Figure 1D*, *Figure 5D–G*). However, the oocyte nucleus is not anchored at the dorsal/anterior corner (*Figure 5D–E,H–K*). *gurken* mRNA is also not localised, although this could be an indirect consequence of the failure in nuclear anchoring (*Figure 5H and I*). The germline clone egg chambers complete oogenesis normally and are laid as eggs, but these never develop, probably because the nuclear localisation defect disrupts meiosis or pronuclear fusion. Thus, most dynein/dynactin-dependent RNA transport processes occur normally in the *arp1*[4D2] mutant. The defects in the delivery of *oskar* mRNA to the posterior cortex, the anchoring of the nucleus, and possibly also *gurken* mRNA localisation must therefore reflect a specific function of dynactin that is disrupted by this allele.

To investigate which function of dynactin is affected by the *arp1*[4D2] mutation, we focused on the defect in the posterior localisation of *oskar* mRNA, as this process has already been extensively characterised. In principle, the failure to deliver *oskar* mRNA to the posterior cortex could reflect either a problem with mRNA transport or a defect in the organisation of the microtubules along which the RNA is transported. Both dynein/dynactin and *oskar* mRNA are transported to the posterior by kinesin 1, and it is therefore possible that dynactin plays a role in coupling *oskar* mRNA to kinesin 1. We therefore examined the movements of *oskar* mRNA in living oocytes using the MS2 system for fluorescently labelling RNA in vivo (*Bertrand et al., 1998*; *Forrest and Gavis, 2003*; *Zimyanin et al., 2008*). The speed and direction of movements as well as mobile fraction of *oskar* mRNA in *arp1*[4D2] homozygotes were very similar to wild-type indicating that dynactin is not required for the kinesin-dependent transport of *oskar* mRNA (*Table 1*). Two aspects of the movements were significantly different from normal, however. Firstly, the number of *oskar* mRNA movements was strongly reduced in the region from 0 to 15 μm from the posterior pole, where the movements are most frequent in wild-type (*Figure 6A*). This provides further evidence that the diffuse localisation of *oskar* mRNA in the posterior cytoplasm in the mutant arises from a failure to transport the mRNA all of the way to the posterior cortex. Secondly, the posterior bias in the direction of *oskar* mRNA movements decreased closer to the posterior pole: the bias is 69% ±2.8 in the region 20–30 μm from the posterior pole; 63% ±3.1, 10–20 μm from posterior and 60% ±3.9, in the region 0–10 μm from posterior pole, whereas the directional bias increases towards the posterior in wild-type (63% ±2.9, 20–30 μm from posterior; 66% ±2.3, 10–20 μm from posterior and 71% ±2.5, 0–10 μm from posterior) (*Figure 6B*; *Parton et al., 2011*).

The *arp1*[4D2] mutant phenotype highlights an unresolved question in *oskar* mRNA localisation, which is how the RNA is actually delivered to the posterior cortex. The microtubules in the oocyte

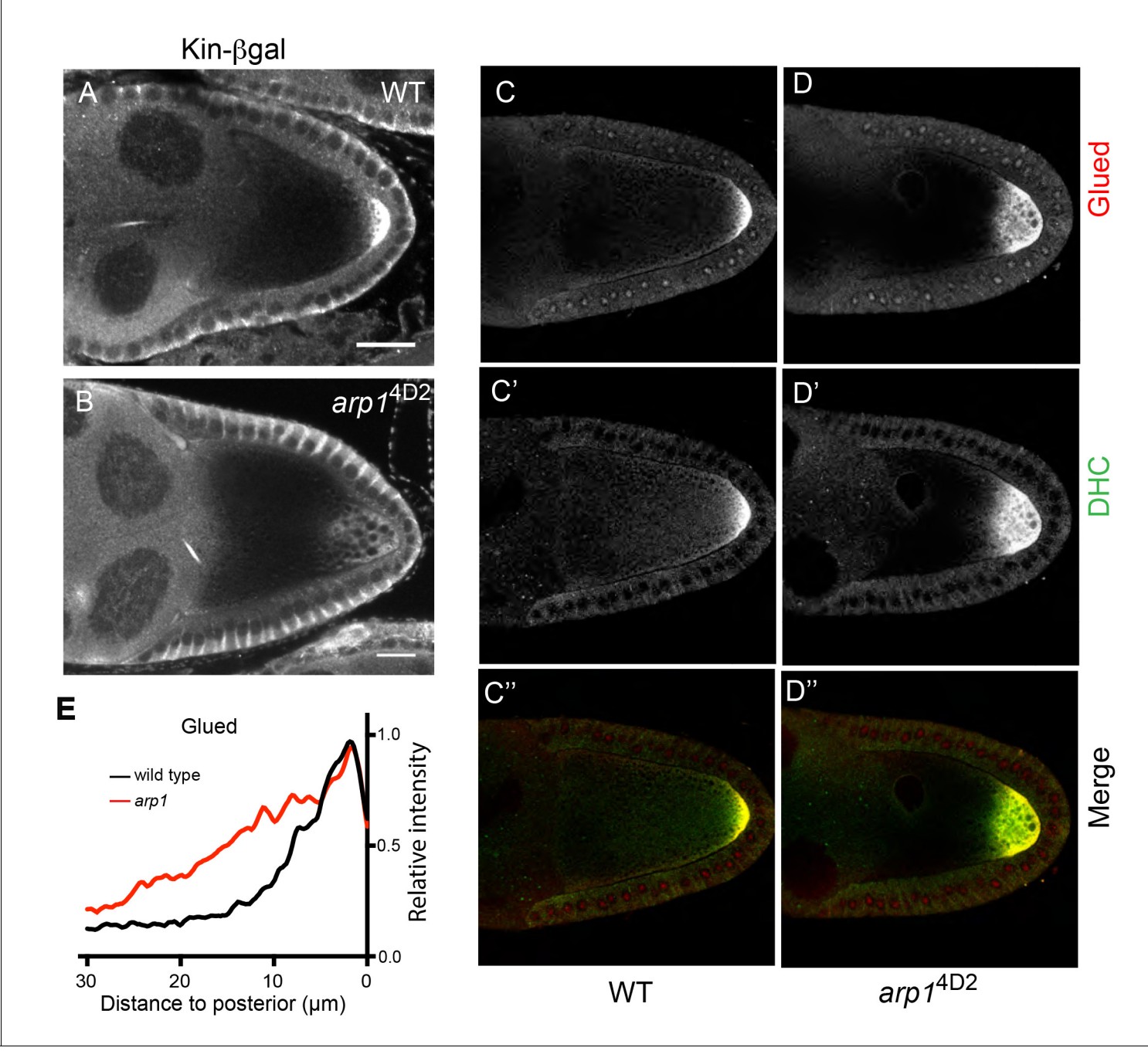

**Figure 3.** *arp1*[4D2] affects the delivery of p150[Glued], the dynein heavy chain and kinesin-βgal to the posterior. (A–B) Wild-type (A) and *arp1*[4D2] mutant (B) stage 9 egg chambers stained for kinesin-βgal. (C–D) Wild-type (C–C'') and *arp1*[4D2] (D–D'') mutant stage 9 egg chambers stained for p150[Glued] (C–D) and the dynein heavy chain (C'–D') The merged image (C''–D'') shows the co-localisation of p150[Glued] (red) and the dynein heavy chain (green). All *arp1*[4D2] mutant oocytes showed the same phenotype (11/11, kinesin-βgal; 19/19, dynein and 16/16, Glued) (E) Average fluorescence intensity profiles in wild type (n = 13) and *arp1*[4D2] mutant (n = 16) egg chambers stained for p150[Glued]. p150[Glued] accumulates in a region extending 15 µm from the posterior cortex in wild type oocytes, but spreads up to 30 µm away from the posterior cortex in *arp1*[4D2] mutant oocytes (p<0.0001 by Wilcoxon Rank Sum Test). Scale bar 20 µm.

DOI: https://doi.org/10.7554/eLife.27237.005

The following figure supplements are available for figure 3:

**Figure supplement 1.** Mapping 4D2 to the Arp1 region.

DOI: https://doi.org/10.7554/eLife.27237.006

*Figure 3 continued on next page*

*Figure 3 continued*

**Figure supplement 2.** Transheterozygous combinations of *arp1* alleles disrupt the localisation of GFP-Staufen to the posterior cortex.

DOI: https://doi.org/10.7554/eLife.27237.007

are highly dynamic and disappear within a few minutes of the addition of drugs, such as colchicine, that sequester free tubulin dimers. Thus, growing microtubules are likely to contact the cortex for only a few seconds before they undergo catastrophe and start shrinking towards the anterior. To deliver *oskar* mRNA to the posterior pole, *oskar* mRNA/Kinesin 1 complexes must therefore reach the plus end of the microtubule while it is still in contact with the posterior cortex (***Figure 6C*** left). One way that dynactin might contribute to this process is by preventing *oskar* mRNA/kinesin complexes from running off the ends of growing microtubules by tethering the complex to the growing plus ends through p150^Glued (***Figure 6C*** right). In wild-type, multiple *oskar* mRNA/kinesin complexes could then track the growing plus ends and be deposited on the cortex when the microtubule reaches the posterior, whereas these complexes would fall off the end of the microtubule in the *arp1*^4D2 mutant. If this model is correct, one would expect a proportion of *oskar* mRNA particles to move at the speed of the growing microtubules in wild-type. However, there is no significant peak in the distribution of *oskar* mRNA velocities that corresponds to the speed of the growing microtubule plus ends, indicating that dynactin does not couple the RNA particles to the plus ends (***Figure 6D***; *Trovisco et al., 2016*). Since kinesin 1 transports *oskar* mRNA with an average speed of 0.47 µm/ sec, while the microtubules grow at 0.23 µm/sec, the *oskar* mRNA/kinesin 1 complexes will continually catch up with the growing plus ends and then fall off (***Figure 6D***). The amount of *oskar* mRNA deposited on the posterior therefore primarily depends on the amount of time that the plus ends persist once they reach the posterior cortex.

Since the *arp1*^4D2 mutant does not appear to affect the behaviour of *oskar* mRNA directly, we next examined the organisation of the microtubules. Anti-tubulin stainings of fixed egg chambers showed a slight reduction in the density of microtubules in *arp1*^4D2 homozygotes compared to wild-type, but no significant change in their overall arrangement (data not shown). However, these stainings do not preserve the microtubules near the posterior of the oocyte. We therefore examined the behaviour of the growing microtubule plus ends by making movies of oocytes expressing the plus end tracking protein EB1 fused to GFP (*Parton et al., 2011*). In wild-type, the growing plus ends slow down close to the posterior to a speed of 0.18 µm/sec, but this does not occur in *arp1*^4D2 homozygotes and the microtubules continue to grow at 0.24 µm/sec (***Figure 7A***). More importantly, very few plus ends extend into the most posterior region of the oocyte (***Figure 7B and C***). Quantifying the frequency of EB1 tracks in this region reveals that almost all microtubules stop in the region between 10 and 20 µm from the posterior cortex in the mutant, with only a very few growing to the posterior pole (***Figure 7D***). To confirm that this was due to an increased catastrophe rate in the *arp1*^4D2 mutant, we measured the lifespan of EB1 comets in the posterior region (***Figure 7E***). Comets can disappear either because the microtubule undergoes catastrophe or because its plus end moves out of the imaging plane. Since the probability of a microtubule moving out of the imaging plane increases the longer the microtubule grows, the apparent lifespan of longer comets is underestimated. Nevertheless, this analysis revealed a significant reduction in the lifespan of EB1 comets in *arp1*^4D2 homozygotes: the mean persistence time of comets in wild-type was 11.29 s (SEM = 0.14, n = 2058) compared to 8.94 s in *arp1*^4D2 homozygotes (SEM = 0.39, n = 211; p<0.0001 by the unpaired t test.). Thus, the *arp1*^4D2 mutation increases both the growth rate and the catastrophe rate of the microtubule plus ends near the posterior. This indicates that wild-type dynactin, which is highly concentrated in the posterior region, acts as an anti-catastrophe factor that allows the growing plus ends to extend all of the way to the posterior pole.

These results suggest that the diffuse localisation of *oskar* mRNA in the posterior cytoplasm in the *arp1*^4D2 mutant is a consequence of the microtubules being too short. To examine whether this effect is sufficient to account for the mutant phenotype, we used the computer simulations of microtubule organisation, cytoplasmic flows and *oskar* mRNA transport to test the effects of varying the microtubule length parameter (*Khuc Trong et al., 2015*). When the microtubules have a mean target length of 0.5 x the distance from the anterior to the posterior of the oocyte (~25 µm), the simulation reproduces the wild-type microtubule organisation and the robust posterior localisation of *oskar* mRNA (***Figure 7F***, left). Shortening the microtubule mean target length by 30% to 17.5 µm still

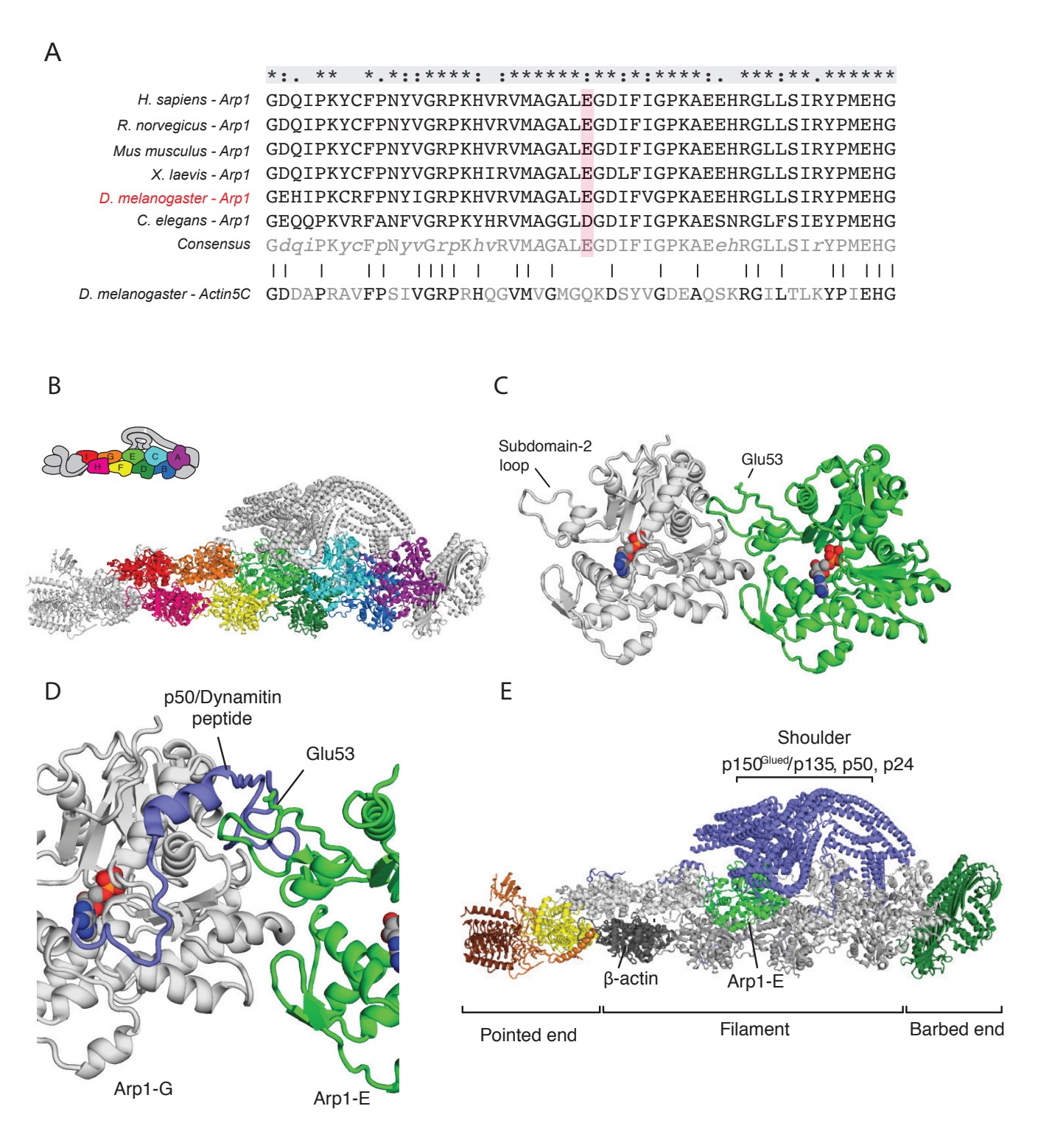

**Figure 4.** The *arp1*[4D2] mutation is predicted to disrupt the interaction between the Arp1 rod and the p150[Glued]/dynamitin/p24 shoulder domain in the Dynactin complex. (**A**) A ClustalX alignment of part of subdomain 2 (amino acids 26–78) in the Arp1 orthologues from different animal species and *Drosophila* Actin 5C. The orthologues shown are *Homo sapiens* (human), *Rattus norvegicus* (rat), *Canis familiaris* (dog), *Xenopus leavis* (African clawed toad), *Drosophila melanogaster* and *Caenorhabditis elegans*. The pink shading marks the conserved glutamate 53, which is mutated to lysine in the *arp1*[4D2] mutant. (**B**) The structure of dynactin, as determined by Cryo-EM. Eight Arp1 subunits (A, B, C, D, E, F, G, and I) and one actin molecule (**H**)

*Figure 4 continued on next page*

*Figure 4 continued*

assemble into two protofilaments that form the rod along the backbone of dynactin. The positions of each subunit are labelled in the drawing at the top and are shown in the same colour in the structure below. (C) A high resolution view of the interface between two Arp1 subunits, showing that E53 does not contribute to the interaction interface, but protrudes laterally. (D) A high resolution view of the interface between Arp1-E subunit and one of the extended peptide domains of p50 dynamitin that anchors the shoulder domain to the Arp1 rod. Glutamate 53 lies at this interface and its substitution by lysine is predicted to disrupt this interaction. (E) A view of the entire dynactin structure showing the positions of Arp1-E (light green) and the p150[Glued] shoulder domain (blue).

DOI: https://doi.org/10.7554/eLife.27237.008

results in the posterior enrichment of *oskar* mRNA, but in a cloud in the posterior cytoplasm rather than at the cortex, exactly as observed in the *arp1*[4D2] mutant (*Figure 7F*, right). Thus, the simulations support the view that the *arp1*[4D2] phenotype is caused by the reduced length of the posterior microtubules.

These results raise the question of whether the kinesin-dependent transport of dynactin contributes to its anti-catastrophe function by delivering dynactin to the growing microtubule plus ends and concentrating it posteriorly. To investigate this issue, we examined the behaviour of growing microtubule plus ends labelled with EB1-GFP in germline clones of *khc*[27], a null mutant in the kinesin heavy chain. As seen in the *arp1*[4D2] mutant, the microtubules grow more rapidly in *khc*[27] homozygous oocytes than in wild-type (*Figure 7—figure supplement 1A*). Furthermore, the relative frequency of EB1 comets is significantly reduced in the region within 20 μm of the posterior cortex in the mutant (p<0.0001) and the lifespan of the comets is also reduced from a median of 9.01 s in wild-type to 7.74 s in *khc*[27] homozygotes (p=0.0006, Wilcoxon rank-sum test). Thus, a failure to transport dynactin to the growing microtubule plus ends near the posterior of the oocyte has a similar effect to the *arp1* mutation, presumably because dynactin needs to be concentrated in this posterior region to efficiently bind the growing plus ends and suppress microtubule catastrophes.

The final issue that we addressed is whether the anti-catastrophe function of dynactin continues when the microtubules reach the posterior cortex, as this would give more time for kinesin 1 to deliver *oskar* mRNA to its anchoring site. EB1 recognises the GTP-bound tubulin incorporated into the growing plus end, while increasing the hydrolysis of GTP to GDP (*Maurer et al., 2011*; *Maurer et al., 2012*; *Seetapun et al., 2012*; *Maurer et al., 2014*; *Zhang et al., 2015*; *Duellberg et al., 2016*). Catastrophe occurs when the EB1-binding GTP-tubulin cap of the microtubule is reduced below a critical threshold, triggering a rapid loss of tubulin dimers from this end. EB1 comets have been previously observed to slow down to 0.08 μm/sec on hitting the cortex of tissue culture cells and can persist there for a few seconds (*Straube and Merdes, 2007*; *van der Vaart et al., 2013*). Thus, the duration of a 'static' EB1-GFP signal at the cortex provides a readout of how long the microtubule plus end persists before undergoing catastrophe. We plotted kymographs along regions of either the lateral or the posterior cortex and measured the lifetime of the static EB1 foci (*Figure 8A–C*). These foci, which appear as vertical lines in the kymograph, persist significantly longer at the posterior cortex than at the lateral cortex. Quantifying this effect reveals that microtubules remain for an average of 15 s at the posterior, with a maximum of over 50 s, compared to a mean persistence time of only 8 s laterally (p=4.1 $\times$ 10$^{-11}$; Wilcoxon rank-sum test). Thus, dynactin may continue to protect the plus ends after they have reached the posterior cortex, providing more time for kinesin 1 to transport *oskar* mRNA to the posterior pole (*Figure 8D*).

## Discussion

Because dynactin and dynein are required for oocyte determination, their functions later in oogenesis have been studied by over-expressing p50 dynamitin or shRNAs against the dynein heavy chain (*Duncan and Warrior, 2002*; *Januschke et al., 2002*; *Sanghavi et al., 2013*). Both treatments disrupt the localisation of *bicoid* mRNA and the anchoring of the oocyte nucleus, but only slightly reduce the amount of *oskar* mRNA that is correctly localised to the posterior cortex. This may be an indirect consequence of reduced transport of *oskar* mRNA from the nurse cells into the oocyte, as these treatments merely lower the levels of wild-type dynactin or dynein (*Clark et al., 2007*; *Mische et al., 2007*). Here, we report a very different phenotype when the entire germline is homozygous for the E53K mutation in the Arp1 subunit of dynactin: oocyte determination, mRNA

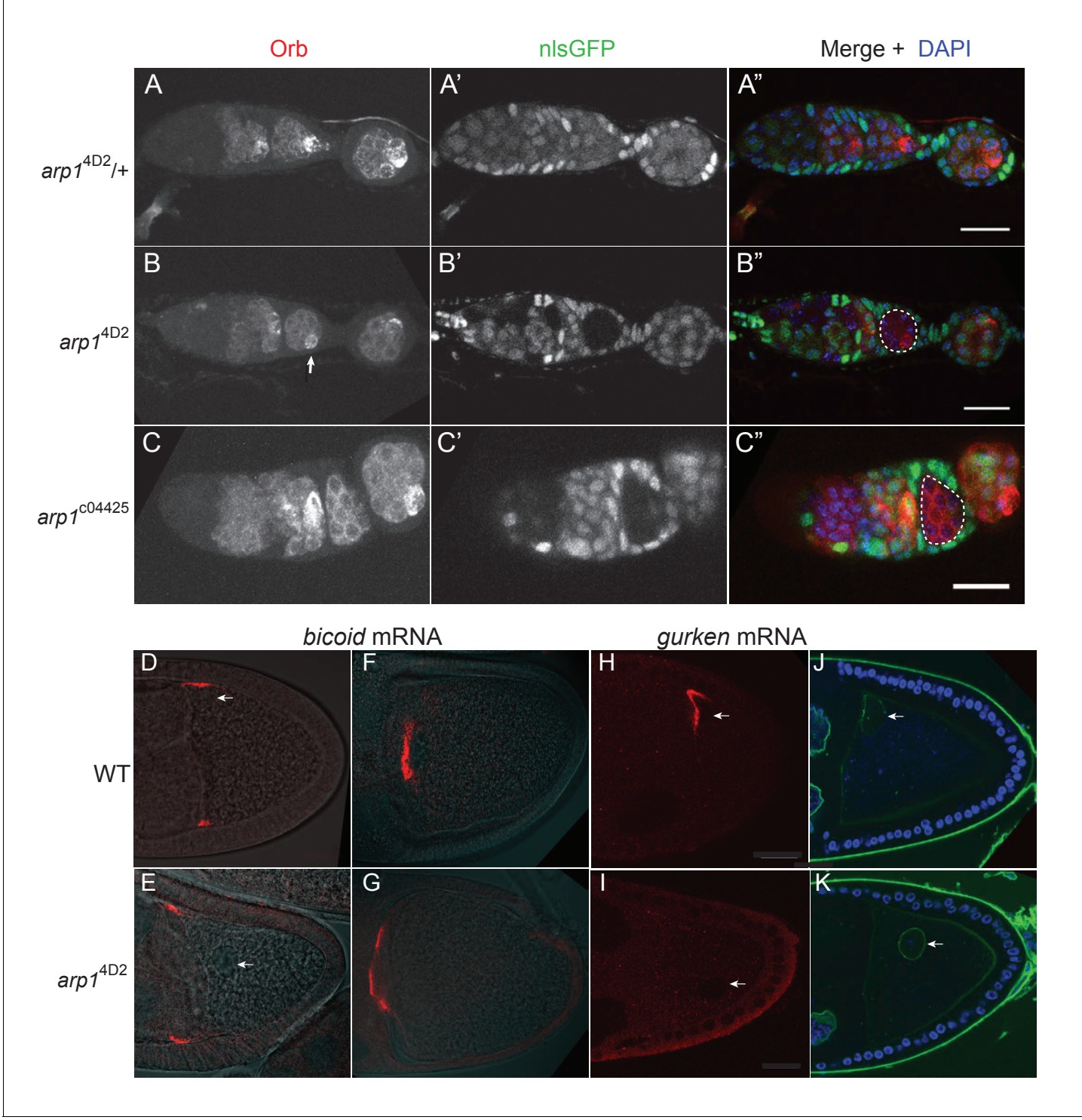

**Figure 5.** The *arp1*[4D2] allele does not disrupt oocyte determination, mRNA transport from the nurse cells to the oocyte or *bicoid* mRNA localisation. (A–A") A confocal image of a germarium from an *arp1*[4D2]/+heterozygote stained for the oocyte marker Orb (white in A; red in A') and expressing nls-GFP (white in A'; green in A'). (B–B") A confocal image of a germarium containing *arp1*[4D2] homozygous germline cysts marked by the loss of nls-GFP (B'). The oocyte has been specified normally in the *arp1*[4D2] mutant germline cyst in region 3 (indicated by the white dashed circle in B'), as shown by the strong enrichment of Orb in one cell (white arrow in B). (C–C") A confocal image of a germarium containing germline clones of the null mutation, *arp1*[c04425]. Loss of Arp1 prevents oocyte specification, as shown by the uniform distribution of Orb in the homozygous mutant cyst in region 2b (indicated by the white dashed circle in C'). Scale bar 20 µm. (D–E) Confocal images showing the localisation of *bicoid* mRNA at stage 9 in a wild-type egg chamber (D) and an *arp1*[4D2] germline clone. The *bicoid* mRNA FISH signal (red) is superimposed on a greyscale DIC image of each egg chamber

*Figure 5 continued on next page*

Figure 5 continued

that shows the positions of the cells. *bicoid* mRNA localises normally to the anterior/lateral corners of the oocyte in the *arp1*[4D2] germline clone. White arrows point to the nucleus. 15/15 stage 9 egg chambers mutant for *arp1*[4D2] showed normal *bicoid* mRNA localisation. (F–G) Confocal images showing the localisation of *bicoid* mRNA at stage 10b in a wild-type egg chamber (F) and an *arp1*[4D2] germline clone (G). *bicoid* mRNA has relocalised to a disc in the centre of the anterior cortex in the *arp1*[4D2] germline clone as in wild-type. 11/11 stage 10 egg chambers mutant for *arp1*[4D2] showed normal *bicoid* mRNA localisation. (H–I) Confocal images showing the localisation of *gurken* mRNA (red; FISH) at stage 9 in a wild-type egg chamber (H) and an *arp1*[4D2] germline clone (I). *gurken* mRNA localises above the oocyte nucleus in the dorsal anterior corner of the oocyte in wild-type, but is not localised in the *arp1*[4D2] mutant, which also disrupts the anchoring of the nucleus (white arrow). 14/21 stage nine egg chambers mutant for *arp1*[4D2] showed similar *gurken* mRNA mislocalisation. (J–K) Confocal images of stage 9 wild-type (J) and *arp1*[4D2] germline clone (K) egg chambers in which the nuclear envelope is stained with wheat germ agglutinin (green) and DNA with DAPI (blue). The *arp1*[4D2] mutant disrupts the anchoring of the nucleus (white arrow) at the dorsal/anterior corner of the oocyte, and it is instead found at random positions within the oocyte cytoplasm. 30/44 stage 9 egg chambers and 53/56 stage 10 egg chambers mutant for *arp1*[4D2] showed similar nuclear mislocalisation.

DOI: https://doi.org/10.7554/eLife.27237.009

transport into the oocyte and *bicoid* mRNA localisation are unaffected, but *oskar* mRNA is diffusely localised in the posterior cytoplasm, rather than at the posterior cortex. This defect is a consequence of an increased catastrophe rate of the microtubules growing towards the posterior, indicating a novel requirement for dynactin in extending microtubules posteriorly, so that kinesin 1 can deliver *oskar* mRNA to its cortical anchoring site.

Dynactin has been shown to function as an anti-catastrophe factor in mammalian tissue culture cells and in neurons, where it promotes the growth of microtubules towards the axonal terminals (*Komarova et al., 2002*; *Lazarus et al., 2013*). This activity depends on the neuronal isoforms of p150[Glued], which contain both an N-terminal CAP-Gly domain and an internal basic region. *Drosophila* p150[Glued] is not alternatively spliced and contains well-conserved CAP-Gly and basic domains, making it a good candidate for the subunit that mediates the anti-catastrophe function of *Drosophila* dynactin in the oocyte. The *arp1*[4D2] mutation is predicted to disrupt the interaction between the Arp1 rod and the dynactin shoulder domain that contains p150[Glued], and this may disturb an allosteric interaction that is necessary for the correct conformation of the p150[Glued] N-terminus. The association of dynactin with growing plus ends is thought to depend on the interaction of p150[Glued] with the C-terminus of CLIP-170 (CLIP-190 in *Drosophila*), which in turn binds to the tail of EB1 (*Duellberg et al., 2014*; *Honnappa et al., 2006*; *Lansbergen et al., 2004*). However, CLIP-190 null mutations are viable and fertile and have no discernible effect on *oskar* mRNA localisation (*Dix et al., 2013*; data not shown). Thus dynactin must be able to associate with the growing plus ends independently of CLIP-190, presumably through its interaction with EB1 (*Duellberg et al., 2014*; *Askham et al., 2002*; *Komarova et al., 2002*; *Ligon et al., 2003*; *Vaughan et al., 2002*). Consistent with our results with *arp1*[4D2], deletion of p150[Glued] N-terminus has no effect on dynein-

**Table 1.** Quantification of oskar mRNA particle movements.

| Phenotype | Average speed, μm/s | Tracks to Posterior, % | Mobile fraction, % |
|---|---|---|---|
| WT | 0.36 ± 0.01 (N=1075) | 65 ± 0.8 (N=989) | 12.0 ± 2.1 (N = 358) |
| arp1[4D2] | 0.37 ± 0.03 (N=296) | 62 ± 0.7 (N=353) | 15.4 ± 2.5 (N = 361) |

Values shown are means plus and minus the SEM. In all cases, the differences between wild-type and the *arp1*[4D2] mutant were statistically insignificant, p>0.05.

DOI: https://doi.org/10.7554/eLife.27237.010

The following source data available for Table 1:

**Source data 1.** Speed of oskar mRNA particles.
DOI: https://doi.org/10.7554/eLife.27237.011

**Source data 2.** Direction of oskar mRNA particles.
DOI: https://doi.org/10.7554/eLife.27237.012

**Source data 3.** Mobile fraction of oskar mRNA particles.
DOI: https://doi.org/10.7554/eLife.27237.013

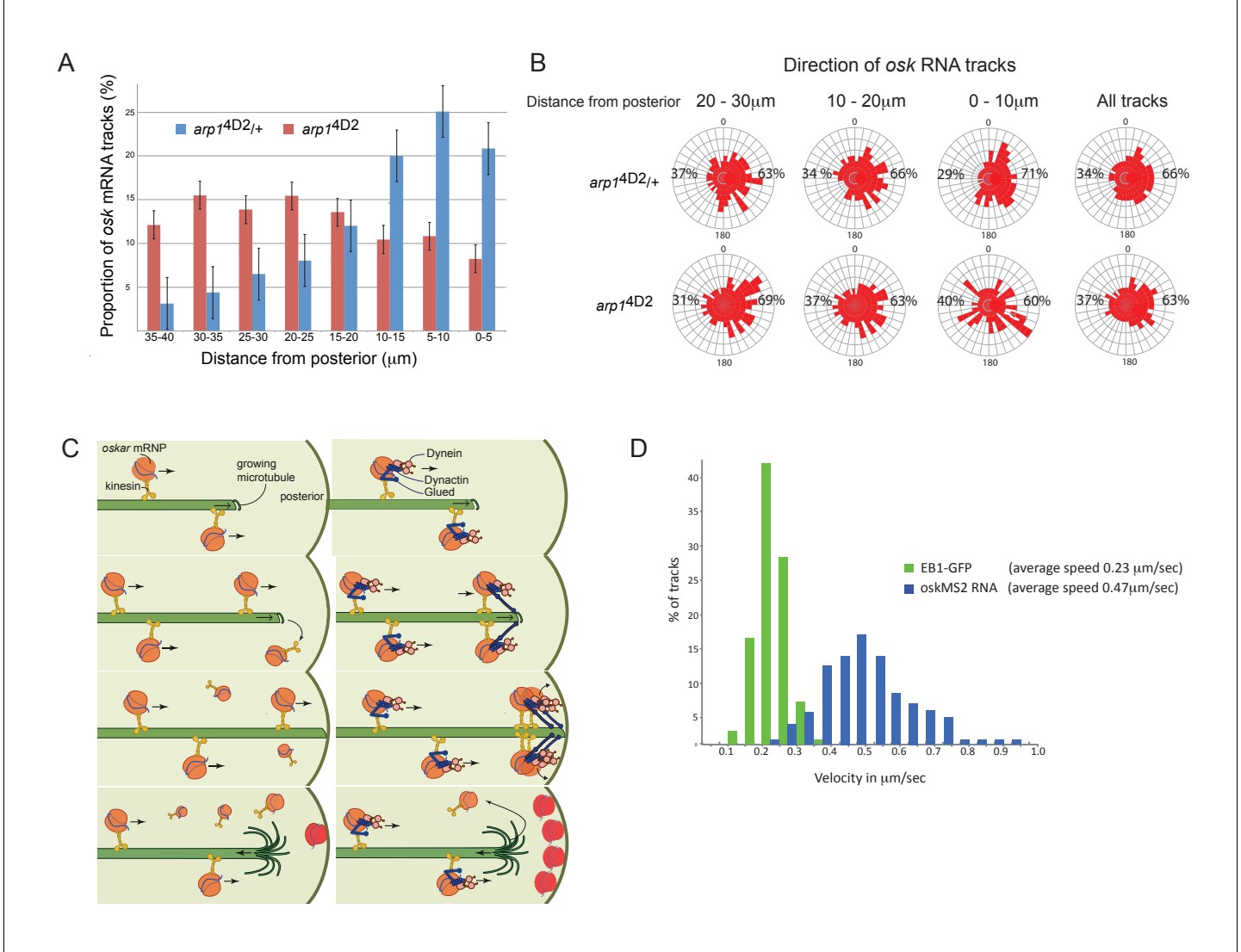

**Figure 6.** The *arp1*[4D2] mutation reduces the frequency and the directional bias of *oskar* mRNA movements near the posterior of the oocyte. (**A**) A graph showing the relative frequency of *oskar* mRNA movements at different distances from the posterior pole in *arp1*[4D2]/+ (n = 528 tracks) and *arp1*[4D2] mutant oocytes (n = 410). The posterior pole is set as 0 μm and the movements have been binned into 5 μm intervals. There is a strong reduction in the frequency of *oskar* mRNA movements near to the posterior pole in the *arp1*[4D2] mutant compared to wild-type (p<0.001 by the Wilcoxon rank-sum test). Error bars indicate the SD. (**B**) A circular histogram showing the frequency of *oskar* mRNA tracks in different directions measured in 10° intervals at the specified distances from the posterior pole. The posterior bias in the direction of *oskar* mRNA movement increases with proximity to the posterior (n = 1252 tracks), but decreases in the *arp1*[4D2] mutant (n = 889 tracks). (**C**) A diagram showing two models for *oskar* mRNA delivery to the posterior cortex. In the left model, *oskar* mRNA/kinesin 1 particles move along microtubules until they reach the plus end, where they fall off. *oskar* mRNA can therefore only be delivered to the posterior cortex if kinesin reaches the growing plus end of the microtubule at the same time that the microtubule reaches the cortex. In the right model, *oskar* mRNA/kinesin 1 particles that reach the end of the microtubule remain tethered to the growing plus end by dynactin, through the +TIP activity of p150[Glued]. *oskar* mRNPs therefore accumulate at the growing plus end until the microtubule reaches the posterior cortex, where they can be offloaded and anchored. (**D**) A histogram showing the velocity distributions of growing microtubule plus ends marked by EB1-GFP (green) and *oskar* mRNPs (blue). The absence of a significant sub-population of *oskar* mRNPs that move at the same speed as microtubule growth argues against the model in which the RNA tracks the growing microtubule ends.

DOI: https://doi.org/10.7554/eLife.27237.014

The following source data is available for figure 6:

**Source data 1.** The number of oskar mRNA tracks at the specified distances from the posterior pole.

DOI: https://doi.org/10.7554/eLife.27237.015

**Source data 2.** Direction of oskar mRNA tracks at the specified distances from the posterior pole.

DOI: https://doi.org/10.7554/eLife.27237.016

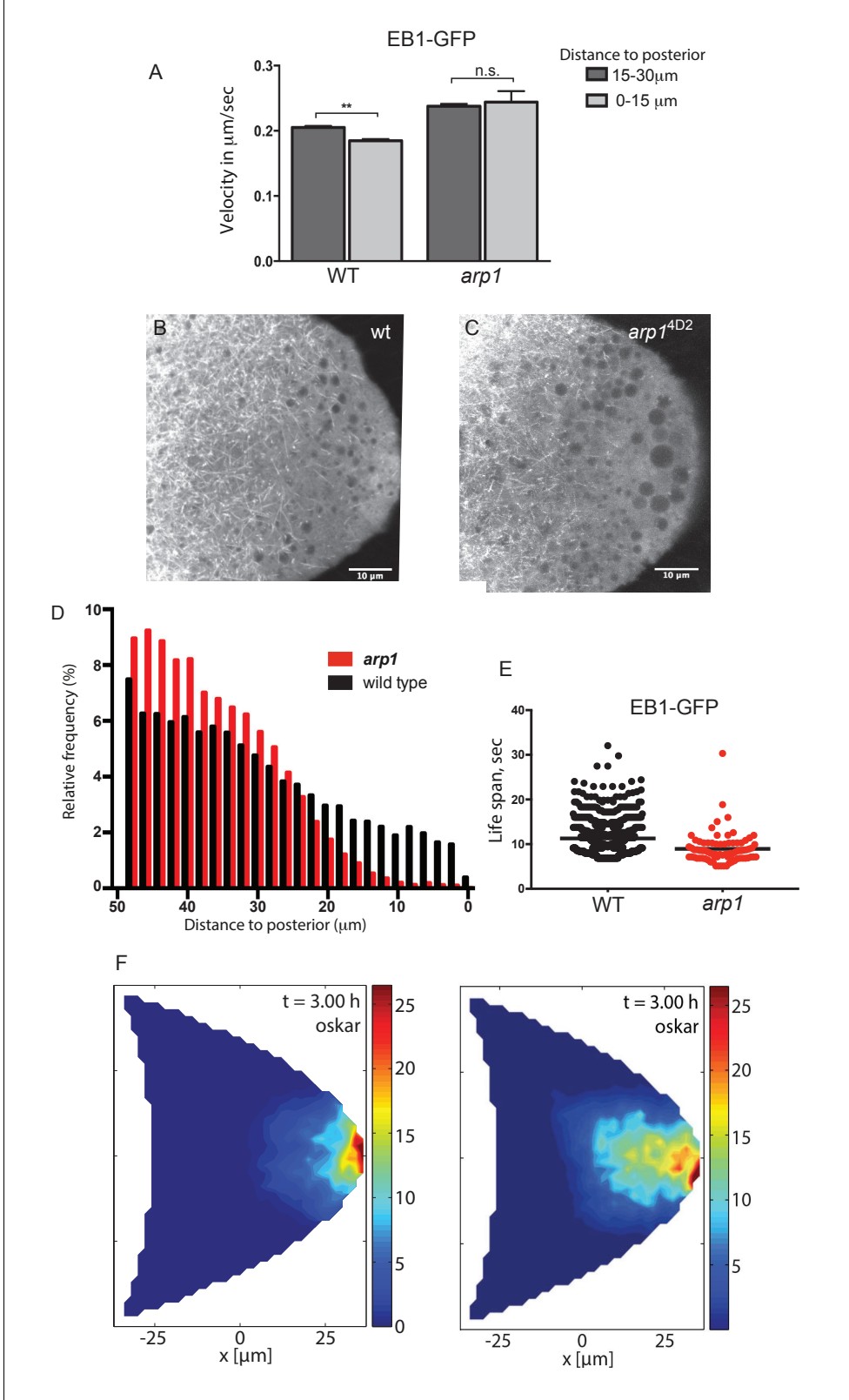

**Figure 7.** Dynactin is required for microtubule growth to the posterior cortex. (**A**) A histogram showing the average velocities (μm/sec) of EB1 comets on growing microtubule plus ends in the centre of the oocyte (15–30 μm from the posterior) and near the posterior cortex (0–15 μm from the cortex) in wild-type (*arp1*$^{4D2}$/+) and *arp1*$^{4D2}$ homozygous oocytes. *arp1*$^{4D2}$/+: 3732 comets in the 15–30 μm region and 2068 comets in 0–15 μm region; *arp1*$^{4D2}$ homozygotes: 2815 comets in the 15–30 μm region and 221 comets in the 0–15 μm region. The plus ends slow down as they approach the posterior
*Figure 7 continued on next page*

*Figure 7 continued*

cortex in wild-type (p=0.007 by the paired t test), but fail to slow down in *arp1*[4D2], suggesting that the dynactin complex normally restricts the rate of microtubule growth in the posterior cytoplasm. Error bars indicate the SEM. (**B–C**) EB1 comet tracks in wild-type (**B**) and *arp1*[4D2] homozygous (**C**) oocytes. The images are merges of 40 frames from time-lapse movies taken at 1.7 s per frame. The tracks therefore represent the growth of microtubule plus ends over 68 s. (**D**) A histogram showing the relative frequency of EB1 comets at different distances from the posterior cortex in wild-type (black; *arp1*[4D2]/+) and *arp1*[4D2] homozygous (red) oocytes. Very few microtubules extend within 10 µm of the posterior cortex in the *arp1*[4D2] mutant (n = 14440 for WT, n = 13062 for *arp1*[4D2], p<0.001 by the Wilcoxon rank-sum test). (**E**) A plot showing the lifespan of growing microtubule plus ends labelled by EB1-GFP near the posterior cortex (0–15 µm from the posterior) in wild type (n = 2058, mean = 11.29, SEM = 0.14) and *arp1*[4D2] homozygous oocytes (n = 211, mean = 8.94, SEM = 0.39). p<0.0001 by the unpaired *t* test. (**F**) Computer simulations of *oskar* mRNA transport with diffusion, motor-transport and cytoplasmic flows, showing the distribution of total cargo after three hours in cross section. The left hand panel (reproduced from *Figure 3F* of *Khuc Trong et al. (2015)* is a simulation in which the mean target length of the microtubules ($\varepsilon$) is set to 0.5 x the anterior-posterior length of the oocyte. The right panel shows an identical simulation in which the mean target length of the microtubules is reduced to 0.35. The shortening of the microtubules changes the simulated distribution of *oskar* mRNA from a tight posterior crescent to a more diffuse posterior cloud.

DOI: https://doi.org/10.7554/eLife.27237.017

The following source data and figure supplement are available for figure 7:

**Source data 1.** Velocities of EB1-GFP comets at the specified distances from the posterior pole.

DOI: https://doi.org/10.7554/eLife.27237.019

**Source data 2.** Distances of EB1-GFP tracks from the posterior cortex.

DOI: https://doi.org/10.7554/eLife.27237.020

**Source data 3.** Lifespan of EB1-GFP comets near the posterior cortex.

DOI: https://doi.org/10.7554/eLife.27237.021

**Figure supplement 1.** Kinesin 1 is required for microtubule growth to the posterior cortex.

DOI: https://doi.org/10.7554/eLife.27237.018

dependent cargo transport, but affects microtubule organisation in *Drosophila* S2 cells (*Kim et al., 2007*).

Although it has been known for many years that dynein and dynactin are transported to the posterior of the oocyte by kinesin 1, the functional significance of this localisation has remained unclear (*Li et al., 1994*; *Palacios and St Johnston, 2002*). Our analysis reveals that kinesin-dependent localisation of dynactin creates a positive feedback loop that amplifies the directional bias in microtubule orientation posteriorly and extends microtubule growth to the posterior pole, both of which are essential for the final step in *oskar* mRNA localisation (*Figure 9*). The oocyte microtubules grow with a weak orientation bias towards the posterior that arises from the fact that they emanate from the anterior and lateral cortex, but are repressed posteriorly (*Khuc Trong et al., 2015*; *Nashchekin et al., 2016*). Plus end-directed transport of dynactin by kinesin 1 along these microtubules will therefore concentrate dynactin posteriorly, where it can associate with the growing plus ends and prevent them from undergoing catastrophes. As a consequence, the plus ends of the microtubules growing into this region extend further towards the posterior, amplifying the directional bias in microtubule orientation posteriorly. This effect explains why the orientation bias in *oskar* mRNA movements increases towards the posterior in wild-type oocytes, but decreases in *arp1*[4D2] mutants (*Figure 6B*). The increased length and directional bias of the microtubules at the posterior allows kinesin 1 to transport dynactin even more posteriorly, generating a positive feedback loop that eventually results in a high posterior concentration of dynactin that promotes microtubule growth all of the way to the posterior cortex. This amplification loop therefore generates the posterior microtubules tracks for the kinesin-dependent transport of *oskar* mRNA to the posterior cortex.

The dynamic nature of the oocyte microtubules means that in order to localise *oskar* mRNA, kinesin/*oskar* mRNA complexes must reach the end of a microtubule at the posterior cortex before the microtubule undergoes catastrophe. This is aided by the fact that the microtubules persist twice as long at the posterior cortex than at the lateral cortex, suggesting that dynactin continues to suppress catastrophes at the cortex. Microtubules undergo catastrophe when subjected to the pushing forces produced by growing against a barrier, such as the cortex (*Janson et al., 2003*). The longer persistence at the posterior could therefore result from the slower growth of the dynactin-associated plus ends, which would increase the pushing force more slowly.

In many cases where microtubule plus ends make sustained contacts with the cortex, such as during centrosome positioning and spindle orientation, cortical dynein captures the plus ends and

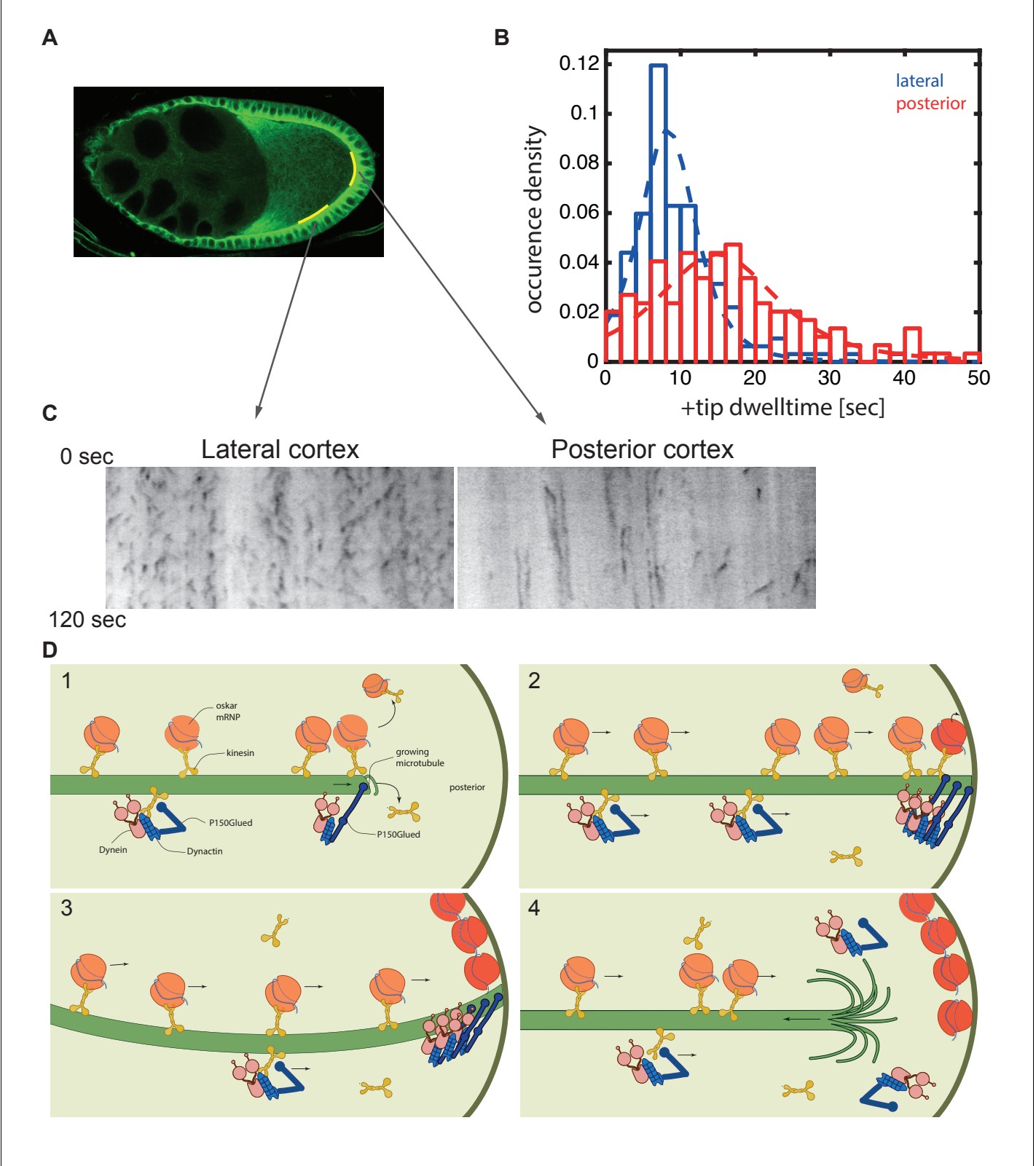

**Figure 8.** Microtubules continue growing for twice as long at the posterior cortex than at the lateral cortex. (**A**) A confocal image of microtubules in a wild-type stage 9 oocyte indicating the regions of the posterior and lateral cortex (yellow lines) from which the kymographs in (**C**) were collected. (**B**) A histogram showing the relative frequency of microtubule plus ends (marked by EB1-GFP) at the lateral (blue, n = 159) and posterior (red, n = 148) cortex of the oocyte as a function of their dwell time. (**C**) Kymographs showing the EB1-GFP signal (black) along regions of the lateral and posterior

*Figure 8 continued on next page*

*Figure 8 continued*

cortex of the oocyte over a period of 120 s. Static EB1 comets result in vertical lines in the kymographs and represent the plus ends of microtubules that have reached the cortex, but have not yet undergone catastrophe. The length of each vertical line therefore represents the lifetime of the growing microtubule at the cortex. The diagonal EB1–GFP tracks in the lateral kymograph represent microtubules that are growing parallel to the cortex from cortical noncentrosomal microtubule organising centres (ncMTOCs). The diagonal tracks are largely absent from the kymograph of the posterior cortex because it lacks ncMTOCs. (D) A series of diagrams showing *oskar* mRNA/kinesin 1 complexes and dynein/dynactin/kinesin1 complexes moving along a microtubule that has Dynactin protecting its growing plus end (panel 1). The dynein/dynactin/kinesin1 complex is shown as different from the *oskar* mRNA/kinesin1 complex because the transport of dynein and dynactin is independent of *oskar* mRNA localisation (*Palacios and St Johnston, 2002*). The microtubule continues to grow upon reaching the posterior cortex (panels 2 and 3), allowing multiple *oskar* mRNA/kinesin1 complexes to move to the plus end and offload onto the cortex, before the microtubule finally undergoes catastrophe and shortens (panel 4).

DOI: https://doi.org/10.7554/eLife.27237.022

The following source data is available for figure 8:

**Source data 1.** Dwell time of EB1-GFP comets at the lateral and posterior cortex

DOI: https://doi.org/10.7554/eLife.27237.023

anchors them to the cortex as they shrink (*Burakov et al., 2003*; *Carminati and Stearns, 1997*; *di Pietro et al., 2016*; *Hendricks et al., 2012*; *Koonce et al., 1999*; *Laan et al., 2012*; *Nguyen-Ngoc et al., 2007*; *Yamamoto et al., 2001*). In budding yeast, for example, a kinesin transports dynein along astral microtubules to the cortex, where dynein is anchored and captures microtubule plus ends to pull on the spindle pole (*Caudron et al., 2008*; *Farkasovsky and Küntzel, 2001*; *Heil-Chapdelaine et al., 2000*; *Markus et al., 2009*; *Sheeman et al., 2003*). Since dynein is transported to the posterior of the oocyte by kinesin 1, and becomes highly enriched in the posterior cytoplasm, like dynactin, it is tempting to speculate that dynein might further extend the time for *oskar* mRNA delivery by tethering shrinking microtubule plus ends to the cortex. However, the very high concentration of dynein in the posterior cytoplasm makes it difficult to tell whether it is specifically recruited to the cortex, and it is not currently possible to image the plus ends of microtubules that have lost their EB1 cap. Thus, investigation of this potential mechanism will require ways to visualise the plus ends of shrinking microtubules.

## Materials and methods

### *Drosophila* stocks and genetics

*white*[1118] was used as a wild-type stock throughout. The following mutant alleles and transgenic lines were used: *arp1*[4D2-12] (*Martin et al., 2003*), *arp1*[1] and *arp1*[2] (*Haghnia et al., 2007*), *arp1*[c04425] (*Thibault et al., 2004*), hsFLP kinβ-Gal (*Clark et al., 1994*), hsFLP maternal α4tubulin::GFP-Staufen (*Martin et al., 2003*), *khc*[27] (*Brendza et al., 2000*), oskMS2 (*Zimyanin et al., 2008*), UAS EB1-GFP (*Jankovics and Brunner, 2006*).

Germline clones were generated with FRT 82B ovoD, FRT 42B ovoD or FRT 82B GFP (Bloomington Stock Center) using the heat shock FLP/FRT system (*Chou and Perrimon, 1992*).

### Immunological/staining methods

Immunofluorescence

Ovaries from 48- to 72-hr-old females were dissected in PBS-T (PBS + 0.2% Tween-20) and fixed in 4% formaldehyde in PBS-T for 20 min. The fixed samples were then incubated with 5% BSA in PBS-T for 1 hr to block nonspecific antibody binding, incubated with primary antibodies in PBS-T plus 1% BSA at 4°C for 18 hr and then washed in PBS-T. If the primary antibody was not directly conjugated to a fluorophore, ovaries were further incubated with fluorophore-conjugated secondary antibodies (for confocal microscopy – 1:200, Jackson Laboratories) and then washed in PBS-T, before addition of Vectashield mounting medium (Vector Laboratories Cat# H-1000 RRID:AB_2336789).

The primary antibodies used were: FITC-conjugated anti-α-tubulin mouse monoclonal (1:100, Sigma-Aldrich, MO, USA, Cat# F2168 RRID:AB_476967), anti-β-galactosidase rabbit polyclonal (1:100, MP Biomedicals, CA, USA, RRID:AB_2335269), anti-Dynein heavy chain mouse monoclonal (1:50, DSHB, RRID:AB_2091523), anti-Glued rabbit polyclonal (1:100, *Nashchekin et al., 2016*), anti-Orb mouse monoclonal (1:200, DSHB, RRID:AB_528418, AB_528419), anti-Oskar guinea pig

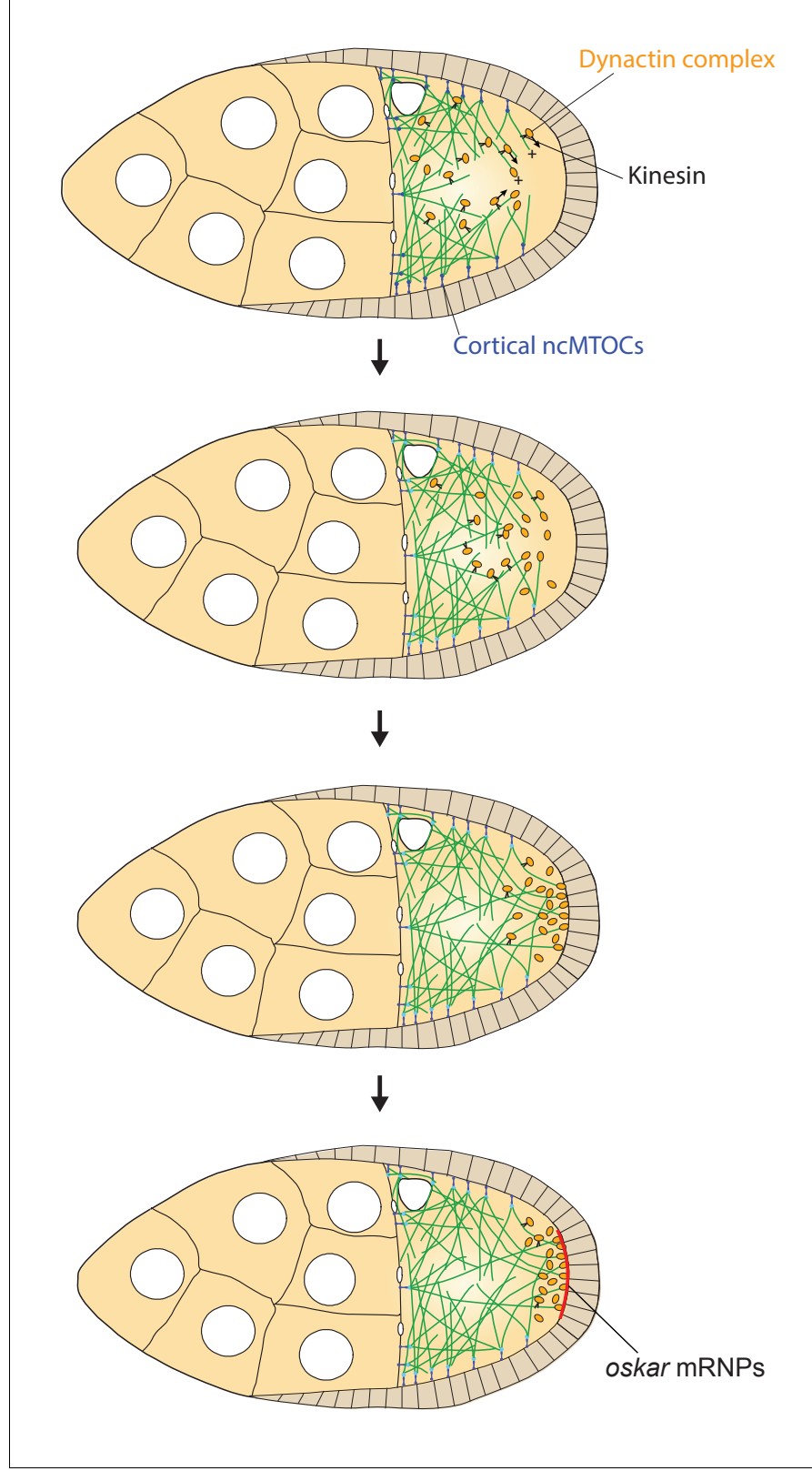

**Figure 9.** A model of the kinesin1/dynactin/microtubule positive feedback loop that increases the length and directional bias of microtubules near the posterior cortex.  A series of diagrams showing the steps in the positive feedback loop that ensures that microtubules reach the posterior cortex to deliver *oskar* mRNA. The distribution of cortical ncMTOCs along the anterior and lateral cortex organizes a weakly polarised microtubule network, with

*Figure 9 continued on next page*

*Figure 9 continued*

more microtubules growing posteriorly than anteriorly. The transport of the dynactin complex along these microtubules by kinesin leads an posterior enrichment of dynactin, where it binds to the growing microtubule plus ends and acts as an anti-catastrophe factor. This causes the microtubules to extend further towards the posterior, which in turn allows kinesin 1 to transport dynactin more posteriorly. This positive feedback eventually results in a high posterior concentration of dynactin and microtubules that extend all of the way to the posterior cortex, thereby allowing the efficient delivery of *oskar* mRNA.

DOI: https://doi.org/10.7554/eLife.27237.024

polyclonal (1:200, *Chang et al., 2011*), anti-Staufen rabbit polyclonal (1:200, *St Johnston et al., 1991*). F-actin was labelled with rhodamine-conjugated Phalloidin (1:200, Thermo Fisher Scientific, MA, USA, Cat# R415). The nuclear membrane was labelled with FITC-conjugated wheat germ agglutinin (1:200, Thermo Fisher Scientific, Cat# W11261).

## Fluorescent in situ hybridisations

Fluorescence in situ hybridisations were performed according to standard protocols. Anti-sense probes for *bcd*, *oskar* and *gurken* RNAs were synthesised using the DIG RNA Labelling mix (Roche, Switzerland) and the linearised plasmids: pGEM_bcd (*Driever et al., 1990*) (cut with BamHI), pBS_osk (cut with Hind III), and pBS_grk (cut with Sal I). Probes were detected with Cy3-conjugated anti- digoxygenin mouse monoclonal antibody (1:200, Jackson Immunoresearch, PA, USA, RRID:AB_2339025)

## Imaging

For live imaging, ovaries were dissected and imaged in Voltalef oil 10S (VWR International) on either a widefield DeltaVision microscope (Applied Precision, WA, USA) equipped with a Photometrics 512 EMCCD camera (Photometrics, AZ, USA) and a 2x magnification tube fitted between the unit and the camera, or an Olympus Fluoview FV1000 confocal microscope (Olympus, Japan) or on an Olympus IX81 inverted microscope with a Yokogawa CSU22 spinning disk confocal imaging system using 100 × 1.4 NA Oil UPlanSApo objective lens (Olympus, Japan). Fixed preparations were imaged using an Olympus Fluoview FV1000 confocal microscope (40 × 1.35 NA Oil UPlanSApo, 60 × 1.35 NA Oil UPlanSApo) or Zeiss LSM510 Meta laser scanning confocal system (Carl Zeiss Microimaging, Inc.) with Plan-Neofloar 40x (Oil) NA1.3 and Plan-Apochromat 63x (Oil) NA1.4 objectives. Images were collected with the softWorXs software (Applied Precision), Olympus Fluoview (Olympus, Japan), MetaMorph Microscopy Automation and Image Analysis Software (Molecular Devices, CA, USA, RRID:SCR_002368) or Zeiss LSM 510 AIM software (Carl Zeiss Microimaging, Inc.) and processed using Fiji (Fiji, RRID:SCR_002285) (*Schindelin et al., 2012*).

## Analysis of *oskar* RNA particles and EB1-GFP comets

Moving oskar-MS2-GFP particles were tracked manually using the MTrackJ plugin for the Fiji image analysis software (Fiji, RRID:SCR_002285) (*Schindelin et al., 2012*). The speed, direction of movement and the mobile fraction of *oskar* RNA particles were analysed as previously described (*Zimyanin et al., 2008*). To visualize oskar-MS2-GFP particles moving near the posterior of the oocyte, we bleached the GFP signal from *oskar* RNA that was already localized and imaged particle movements after 10 min of recovery. We analysed at least 5 oocytes per sample type.

EB1-GFP comets were tracked using plusTipTracker (*Matov et al., 2010*; *Applegate et al., 2011*). For each comet, the speed was calculated as the mean of its velocities at individual time points. Distance from the posterior was measured from its initial position. At least 5 oocytes per sample type were examined. To compute the time growing microtubules persist at the lateral or posterior cortex, individual EB1-GFP tracks were first extracted by the plusTipTracker package. Custom-written MATLAB code (available at GitHub, https://github.com/MaxJakobs/Nieuwburg2017; copy archived at https://github.com/elifesciences-publications/Nieuwburg2017) then separated the tracks into those that extend to the oocyte cortex (excluding those that reach the cortex at angle of <20°). The oocyte cortex was drawn by hand, tracing the fluorescent outline of the cell and all tracks that came within 1 μm of the hand-drawn boundary were considered as touching the cortex.

The cortical dwell time for each track was calculated as the time between the EB1-GFP comet moving within 1 μm of the boundary and its disappearance. Data for 3 different oocytes was pooled, and a location-scaled t-distribution fitted to each posterior and lateral dataset.

The significance of any differences in speed or relative frequency of *oskar* RNA particle and EB1-GFP comet movements was evaluated using t tests and Wilcoxon rank-sum tests. When the distribution was normal, the significance of the difference in the means was assessed using a standard Student's *t* test for distributions with equal variance, or the Welch-corrected Student's *t* test for distributions with unequal variance. When the distribution was non-normal, we compared medians using the non-parametric counterpart of the Student's t test, the Wilcoxon rank-sum test.

## Acknowledgements

We would like to thank Damien Brunner, Larry Goldstein and the Bloomington Stock Center (NIH P40OD018537) for fly stocks, the Gurdon Institute Imaging Facility for assistance with microscopy and Avik Mukherjee for the help with MATLAB. This work was supported by a Wellcome Trust Principal Fellowship to D. St J (080007) and by core support from the Wellcome Trust (092096) and Cancer Research UK (A14492). RN was supported by a bursary from the Island of Jersey government and MJ by a Wellcome Trust 4 year PhD studentship in Developmental Mechanisms (109145).

## Additional information

### Funding

| Funder | Grant reference number | Author |
| --- | --- | --- |
| Wellcome | Principal Fellowship 080007 | Daniel St Johnston |
| Wellcome | PhD studentship 109145 | Maximilian Jakobs |
| Wellcome | core support 092096 | Daniel St Johnston |
| Cancer Research UK | core support A14492 | Daniel St Johnston |
| Island of Jersey govement | bursary | Ross Nieuwburg |

The funders had no role in study design, data collection and interpretation, or the decision to submit the work for publication.

### Author contributions

Ross Nieuwburg, Conceptualization, Resources, Supervision, Funding acquisition, Writing—original draft, Project administration, Writing—review and editing; Dmitry Nashchekin, Conceptualization, Formal analysis, Validation, Investigation, Visualization, Methodology, Writing—review and editing; Maximilian Jakobs, Conceptualization, Formal analysis, Validation, Investigation, Visualization, Methodology, Writing—original draft, Writing—review and editing; Andrew P Carter, Software, Investigation, Writing—review and editing; Philipp Khuc Trong, Investigation, Visualization, Writing—review and editing; Raymond E Goldstein, Investigation, Visualization; Daniel St Johnston, Resources, Supervision, Investigation

### Author ORCIDs

Dmitry Nashchekin (iD) http://orcid.org/0000-0001-7372-0752
Raymond E Goldstein (iD) https://orcid.org/0000-0003-2645-0598
Daniel St Johnston (iD) http://orcid.org/0000-0001-5582-3301

### Decision letter and Author response

Decision letter https://doi.org/10.7554/eLife.27237.026
Author response https://doi.org/10.7554/eLife.27237.027

## Additional files

**Supplementary files**

• Transparent reporting form

DOI: https://doi.org/10.7554/eLife.27237.025

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
