## [Decision Letter]

Thank you for submitting your article "Localised Dynactin protects growing microtubules to deliver *oskar* mRNA to the posterior cortex of the *Drosophila* oocyte" for consideration by *eLife*. Your article has been reviewed by three peer reviewers, and the evaluation has been overseen by Anna Akhmanova as the Senior Editor and Reviewing Editor. The reviewers have opted to remain anonymous.

The reviewers have discussed the reviews with one another and the Reviewing Editor has drafted this decision to help you prepare a revised submission.

Summary:

This manuscript provides support for a novel mechanism that contributes to the targeting of *oskar* mRNA to the posterior cortex of the *Drosophila* oocyte. The authors identify a mutation in the Arp1 component of the dynactin complex due its effect on kinesin-dependent transport of cargo to the posterior of the oocyte. The mutation does not affect oocyte determination or bicoid localization, which require dynein-mediated transport, leading the authors to conclude that the mutation does not affect the role of dynactin in dynein function. The authors propose a role for dynactin in protecting dynamic microtubules at the posterior region of the oocyte from depolymerization, and thus allowing kinesin-1-associated *oskar* mRNPs to reach the posterior cortex. Because kinesin-1 is required for enrichment of dynactin at the posterior cortex, it appears there is a positive feedback loop supporting the lengthening of microtubules near the posterior. This is an original model, which should appeal to those interested in mRNA localization and those interested in how microtubule dynamics contribute to sorting processes in vivo. However, key elements of the model require further support. Furthermore, quantification and statistics should be included for most of the figures in the paper, and "relative" quantifications, which can be misleading, should be removed wherever possible.

Essential revisions:

1) The arguments/evidence that the *arp1* allele leads to increased incidence of microtubule catastrophe needs to be strengthened. The authors make strong statements such as "This defect is a consequence of an increased catastrophe rate of the microtubules growing towards the posterior…" and "Thus, the *arp1* mutation increases both the growth rate and the catastrophe rate of the microtubule plus ends near the posterior". Yet the catastrophe rate is not explored directly. The deduction of an increased catastrophe rate appears to be based in large part on the observation that there are fewer EB1 tracks in the posterior region relative to more anterior regions in the mutant. Could there be relatively fewer nucleation events in the posterior region of the oocyte in the mutant, which could also explain the observed results with EB1? The authors are able to track the velocity of EB1 comets moving through the cytoplasm even though this is a complex, three-dimensional system. They should attempt to track the persistence time of EB1 signals in the posterior cytoplasm in wild-type and *arp1* mutant oocytes, which should provide information on the relative duration of growth phases and thus give insight into the relative frequency of catastrophes.

2) Another key part of the study is the evidence that dynactin localized at the posterior cortex plays a role in stabilizing EB1-bound plus ends, thus creating a feedback loop that promotes delivery of *oskar* to the cortex. The authors should demonstrate that the localisation of dynactin coincides specifically with the 'anti-catastrophe' region from 10-20 μm in the oocyte, and that dynactin is either absent or at lower concentration in this specific region in the *arp1* mutant cells Furthermore, the authors should comment on whether there are precedents for EB1 being bound for several seconds to static microtubules in cortical regions in other systems. An alternative explanation is that EB1 that is not bound to microtubules also has some affinity for p150, which is strongly enriched at the posterior cortex. The interpretation that the EB1 signals at the posterior in Figure 8 report on microtubule plus ends should be confirmed by disruption of microtubules. For example, injection of colchicine should quickly remove these signals if they derive from microtubules.

3) The authors use computer simulations to support the conclusion that the diffuse localization of *oskar* to the cortex in *arp1*^4D2^ mutants results because there are fewer and less stable microtubules that reach the posterior cortex. This conclusion, and the importance of microtubule stabilization would be much stronger if the authors showed that increasing stabilized microtubules at the cortex results in more localized *oskar* than in wild type/untreated oocytes. The authors should examine the effect on *oskar* localization after taxol treatment.

4) Quantifications and statistical analysis need significant improvement, and the data description and presentation should be made more clear.

a) It would be helpful to include a cartoon in Figure 1, to demonstrate the current knowledge and the problem/question that is addressed in this manuscript, as is described in the third paragraph of the Introduction. This way, the reader would be oriented to the transport system, since microtubules are not actually shown until the end of the paper.

b) Figure 1 is described in one sentence, and it is this figure that seems to provide a basis for the entire paper. Therefore, quantification to support the phenotypes shown in these sample images, as well as a cartoon and further explanation, would be helpful.

c) Quantification over multiple images (with statistics), rather than just one selected image in each case, is necessary to differentiate WT and 4D2 results in Figure 2.

d) Figure 3 are presented before Figure 3, which was confusing – the panels should be switched in the figure.

e) As above, quantification over multiple images, and statistics, are required in Figure 3 to demonstrate that there is a Kin-βgal phenotype in the 4D2 germline clones. In addition, statements such as "very similar defect" are too vague – is this the same phenotype as for the *oskar* mRNA? Are the fluorescence distributions/line scans quantitatively the same for both WT and 4D2 in each case? How are they different, and can this be explained with the model proposed here?

f) In Figure 3 the authors demonstrate that dynein and dynactin show "an identical posterior localization phenotype to *oskar* mRNA" in WT and 4D2 germline clones. Here again, quantification over multiple images, and statistics, are required to make this strong statement.

g) It would be helpful to include an accompanying cartoon in Figure 6 to clearly demonstrate the "distance from the posterior" regions as measured here. Further, in the text, it states "the number of *oskar* mRNA movements was strongly reduced in the region from 0-15um from the posterior pole…" However, the y-axis of the graph in Figure 6 presented the relative percentage of tracks, rather than the number of tracks. It would seem important to also present the number of tracks observed in each case, or at least provide statistics to demonstrate that the overall number of tracks per oocyte is unchanged between the two strains, and so the only difference is in the number of movements by location. On the graph in Figure 6, statistically significant differences between the strains (vs. not significant) should be noted.

h) Figure 6: This is a very interesting observation but statistical evaluation (or at least standard deviations) to support the interpretation are missing. This should be rectified.

i) In regard to Figure 7, the authors state that "almost all microtubules undergo catastrophe in the region between 10 and 20 μm from the posterior cortex in the mutant…" However, relative frequency is shown on the y-axis. Using absolute numbers, or at least demonstrating that the total numbers are similar between *arp1* and WT, is important to demonstrate this point. In particular, are the absolute numbers of EB1 tracks per oocyte at position 20 μm statistically unchanged between WT and *arp1*, but then the number of tracks abruptly drops off past 20 μm for the *arp1*, while remaining constant for WT?

j) It is not clear how robust the modeling in 7E is to different lengths of microtubules. It seems that the authors do not have direct evidence that there is a shift of mean microtubule length of the proportions used in the model to simulate the mutant situation. Why was 17.5 microns chosen as the length for the model?

[Editors' note: further revisions were requested prior to acceptance, as described below.]

Thank you for resubmitting your work entitled "Localised Dynactin protects growing microtubules to deliver *oskar* mRNA to the posterior cortex of the *Drosophila* oocyte" for further consideration at *eLife*. Your revised article has been favorably evaluated by Anna Akhmanova as the Senior Editor and Reviewing Editor, and two reviewers.

The manuscript has been improved but there are some remaining issues that need to be addressed before acceptance, as outlined below. Most importantly, there appears to be a mismatch between the overall mechanistic model put forward in the paper and the data shown in Figure 3, and this will need to be addressed.

Major concerns:

1) The authors propose a model, in which *arp1*^4D2^ disrupts the delivery of p150^Glued^ to the posterior, leading to an increased catastrophe frequency (decreased Eb1-GFP comet frequency) in the range of 0-15 μm from the posterior. While there are no error bars or a statistical analysis of the data in Figure 3, it appears that the relative intensity of p150^Glued^ 0-10 μm from the posterior is nearly identical between WT and mutant, while there may be increased intensity of p150^Glued^ past 10 μm away from the posterior. This plot does not effectively make the argument that *arp1*^4D2^ disrupts delivery of p150^Glued^ to the posterior, since no direct comparison of intensity is provided between WT and mutant. It appears that the data may be normalized to the maximum value in each case, but this just demonstrates that there is a more diffuse signal, rather than that the integrated intensity of p150^Glued^ is higher in WT as compared to mutant in the zone from 0-15 μm from the posterior. In the reviewer response, the authors state that "Even though p150^Glued^ /Dynactin is still present in the "anti catastrophe" area in the mutant, it cannot stabilise microtubule plus ends because of the mutation in Arp1." This statement seems to be at odds with the title of Figure 3, which reads "*arp1*^4D2^ disrupts the delivery of p150^Glued^, the dynein heavy chain and kinesin-βgal to the posterior." More importantly, this statement seems to propose an entirely different model than the primary model that is put forward in the paper. One way of addressing this concern is to quantify the intensity of the signal at the posterior in the wild-type and mutant from several samples per genotype with the same acquisition settings. If the quantification of the data would not support the proposed mechanism, the model might require significant adjustments.

2) Figure 7 would be much stronger if the authors were able to also measure the lifetime of EB1 comets outside of the posterior region – their model is that the catastrophe rate is specifically altered near to the posterior, and so it seems that the lifetimes in Figure 7 should be measured both near to the posterior and away from the posterior – if the effect of the mutant is only at the posterior, then the difference in lifetimes should disappear for measurements away from the posterior.

---

## [Author Response]

Essential revisions:1) The arguments/evidence that the arp1 allele leads to increased incidence of microtubule catastrophe needs to be strengthened. The authors make strong statements such as "This defect is a consequence of an increased catastrophe rate of the microtubules growing towards the posterior…" and "Thus, the arp1 mutation increases both the growth rate and the catastrophe rate of the microtubule plus ends near the posterior". Yet the catastrophe rate is not explored directly. The deduction of an increased catastrophe rate appears to be based in large part on the observation that there are fewer EB1 tracks in the posterior region relative to more anterior regions in the mutant. Could there be relatively fewer nucleation events in the posterior region of the oocyte in the mutant, which could also explain the observed results with EB1? The authors are able to track the velocity of EB1 comets moving through the cytoplasm even though this is a complex, three-dimensional system. They should attempt to track the persistence time of EB1 signals in the posterior cytoplasm in wild-type and arp1 mutant oocytes, which should provide information on the relative duration of growth phases and thus give insight into the relative frequency of catastrophes.

We have now examined the life span of EB1 comets near to the posterior (0-15 μm) and show that in *arp1* mutant oocytes microtubules grow for a significantly shorter period of time compared to wild type oocytes (new Figure 7). We have added such information to the text:

“To confirm that this was due to an increased catastrophe rate in the *arp1*^4D2^ mutant, we measured the lifespan of EB1 comets in the posterior region (Figure 7). […] Nevertheless, this analysis revealed a significant reduction in the lifespan of EB1 comets in *arp1*^4D2^ homozygotes: the mean persistence time of comets in wild-type was 11.29 sec (, SEM=0.14, n=2058) compared to 8.94 sec in *arp1*^4D2^ homozygotes (SEM=0.39, n=211; p<0.0001 by the unpaired t test.).”

2) Another key part of the study is the evidence that dynactin localized at the posterior cortex plays a role in stabilizing EB1-bound plus ends, thus creating a feedback loop that promotes delivery of oskar to the cortex. The authors should demonstrate that the localisation of dynactin coincides specifically with the 'anti-catastrophe' region from 10-20 μm in the oocyte, and that dynactin is either absent or at lower concentration in this specific region in the arp1 mutant cells

We have now carefully measured p150^Glued^ localisation at the posterior and show that its distribution coincides with the “anti-catastrophe” region of 0-15 μm from the posterior in wild type (new Figure 3). However, as we previously showed in Figure 3, p150^Glued^ extends a much greater distance from the posterior in the *arp1* mutant. Even though p150^Glued^ /Dynactin is still present in the “anti catastrophe” area in the mutant, it cannot stabilise microtubule plus ends because of the mutation in Arp1.

Furthermore, the authors should comment on whether there are precedents for EB1 being bound for several seconds to static microtubules in cortical regions in other systems.

EB1 comets have been observed to slow down to 0.08μm/sec at the cell cortex (van der Vaart 2013). We assume that this could also be the case for microtubules at the posterior cortex where microtubules are not necessarily static but could be in a slow growth mode. We have added the following sentence to the text:

“EB1 comets have been previously observed to slow down to 0.08μm/sec on hitting the cortex of tissue culture cells and can persist there for a few seconds (Straube and Merdes, 2007; van der Vaart et al., 2013)”

An alternative explanation is that EB1 that is not bound to microtubules also has some affinity for p150, which is strongly enriched at the posterior cortex. The interpretation that the EB1 signals at the posterior in Figure 8 report on microtubule plus ends should be confirmed by disruption of microtubules. For example, injection of colchicine should quickly remove these signals if they derive from microtubules.

Treatment of the oocyte with colchicine quickly removes EB1 signals from the posterior area (new video for the referees, Zhao et al., 2012) confirming that posterior EB1 signal derives from binding to microtubule plus ends.

3) The authors use computer simulations to support the conclusion that the diffuse localization of oskar to the cortex in arp1^4D2^ mutants results because there are fewer and less stable microtubules that reach the posterior cortex. This conclusion, and the importance of microtubule stabilization would be much stronger if the authors showed that increasing stabilized microtubules at the cortex results in more localized oskar than in wild type/untreated oocytes. The authors should examine the effect on oskar localization after taxol treatment.

Taxol treatment causes microtubule bundling and re-arrangement of oocyte MT network that makes it difficult to interpret the outcome of such an experiment.

4) Quantifications and statistical analysis need significant improvement, and the data description and presentation should be made more clear.a) It would be helpful to include a cartoon in Figure 1, to demonstrate the current knowledge and the problem/question that is addressed in this manuscript, as is described in the third paragraph of the Introduction. This way, the reader would be oriented to the transport system, since microtubules are not actually shown until the end of the paper.

We have added a cartoon (new Figure 1).

b) Figure 1 is described in one sentence, and it is this figure that seems to provide a basis for the entire paper. Therefore, quantification to support the phenotypes shown in these sample images, as well as a cartoon and further explanation, would be helpful.

We have now added quantification of the phenotypes to the legend of Figure 1.

c) Quantification over multiple images (with statistics), rather than just one selected image in each case, is necessary to differentiate WT and 4D2 results in Figure 2.

We have now added quantification of the phenotypes to the legend of Figure 2.

d) Figure 3 are presented before Figure 3, which was confusing – the panels should be switched in the figure.

The Figure 3 has been now rearranged.

e) As above, quantification over multiple images, and statistics, are required in Figure 3 to demonstrate that there is a Kin-βgal phenotype in the 4D2 germline clones.

We have now added quantification of the phenotypes to the legend of Figure 3.

In addition, statements such as "very similar defect" are too vague – is this the same phenotype as for the oskar mRNA? Are the fluorescence distributions/line scans quantitatively the same for both WT and 4D2 in each case? How are they different, and can this be explained with the model proposed here?

We disagree that “very similar defect” is too vague a description. *oskar* RNA, Staufen, kinbGal and Glued all show the same phenotype with 100% penetrance in 4D2 germline clones. This is an unusual and highly specific phenotype of “a diffuse cloud near the posterior cortex”. We do not believe that quantifying the fluorescence distribution will add a great deal, but we have performed such an analysis for p150^Glued^ (new Figure 3).

f) In Figure 3 the authors demonstrate that dynein and dynactin show "an identical posterior localization phenotype to oskar mRNA" in WT and 4D2 germline clones. Here again, quantification over multiple images, and statistics, are required to make this strong statement.

We have now added a quantification of the phenotypes to the legend of Figure 3.

We have measured p150^Glued^ fluorescence intensity in mutant and wild type oocytes (new Figure 3).

g) It would be helpful to include an accompanying cartoon in Figure 6 to clearly demonstrate the "distance from the posterior" regions as measured here.

We believe that distance from posterior is self-explanatory, and cannot easily fit in an extra cartoon.

Further, in the text, it states "the number of oskar mRNA movements was strongly reduced in the region from 0-15um from the posterior pole…" However, the y-axis of the graph in Figure 6 presented the relative percentage of tracks, rather than the number of tracks. It would seem important to also present the number of tracks observed in each case, or at least provide statistics to demonstrate that the overall number of tracks per oocyte is unchanged between the two strains, and so the only difference is in the number of movements by location. On the graph in Figure 6, statistically significant differences between the strains (vs. not significant) should be noted.

We use relative percentage of tracks because the total number of *oskar* mRNA particles varies between oocytes of the same genotype as well as between genotypes. Using the relative frequencies provides a way of comparing across oocytes and genotypes by normalising for the total number of tracks. The exact number of tracks in each region and the statistical significance are provided in the figure legend and in the corresponding supplementary source data file.

h) Figure 6: This is a very interesting observation but statistical evaluation (or at least standard deviations) to support the interpretation are missing. This should be rectified.

We have added such information to the text:

“Secondly, the posterior bias in the direction of *oskar* mRNA movements decreased closer to the posterior pole: the bias is 69% ± 2.8 in the region 20-30 μm from the posterior pole; 63% ± 3.1, 10-20 μm from posterior and 60% ± 3.9, in the region 0-10 μm from posterior pole, whereas the directional bias increases towards the posterior in wild-type (63% ± 2.9, 20-30 μm from posterior; 66% ± 2.3, 10-20 μm from posterior and 71% ± 2.5, 0-10 μm from posterior) (Figure 6; Parton et al., 2011).”

i) In regard to Figure 7, the authors state that "almost all microtubules undergo catastrophe in the region between 10 and 20 μm from the posterior cortex in the mutant…" However, relative frequency is shown on the y-axis. Using absolute numbers, or at least demonstrating that the total numbers are similar between arp1 and WT, is important to demonstrate this point. In particular, are the absolute numbers of EB1 tracks per oocyte at position 20 μm statistically unchanged between WT and arp1, but then the number of tracks abruptly drops off past 20 μm for the arp1, while remaining constant for WT?

We use relative frequencies for the same reason as for *oskar* mRNA tracks. The total number of EB1 comets varies between oocytes of the same genotype and using relative frequencies normalises for these variations. The number of tracks for each oocyte can be found in Figure 7—source data 2.

j) It is not clear how robust the modeling in 7E is to different lengths of microtubules. It seems that the authors do not have direct evidence that there is a shift of mean microtubule length of the proportions used in the model to simulate the mutant situation. Why was 17.5 microns chosen as the length for the model?

We agree that we did not measure microtubule length directly (because this is impossible to do in a such dense meshwork of microtubules) but we have good indirect evidence from studying EB1 dynamics that microtubules are shorter in the *arp1* mutant. The model we developed in Trong et al. reproduces the observed microtubule organisation in the oocyte very well with a mean microtubule length of 25μm (determined by running simulations with different lengths). We therefore used this model to test the effect of reducing microtubule length by 30% (to a mean of 17.5μm in the simulation). This shows that this small reduction in microtubule length disrupts *oskar* RNA localisation to give a distribution that is similar to that observed in *arp1* mutants. We believe that the modelling reinforces our conclusion that microtubule length, particularly at the posterior, is a crucial parameter for RNA localisation, and that this does not depend on the actual numbers used in the model.

To make this point more clearly, we have revised the text to state:

“Shortening the microtubule mean target length by 30% to 17.5μm still results in the posterior enrichment of *oskar* mRNA, but in a cloud in the posterior cytoplasm rather than at the cortex, exactly as observed in the *arp1*^4D2^ mutant (Figure 7, right)”

[Editors' note: further revisions were requested prior to acceptance, as described below.]

The manuscript has been improved but there are some remaining issues that need to be addressed before acceptance, as outlined below. Most importantly, there appears to be a mismatch between the overall mechanistic model put forward in the paper and the data shown in Figure 3, and this will need to be addressed.Major concerns:1) The authors propose a model, in which arp1^4D2^ disrupts the delivery of p150^Glued^ to the posterior, leading to an increased catastrophe frequency (decreased Eb1-GFP comet frequency) in the range of 0-15 μm from the posterior. While there are no error bars or a statistical analysis of the data in Figure 3, it appears that the relative intensity of p150^Glued^ 0-10 μm from the posterior is nearly identical between WT and mutant, while there may be increased intensity of p150^Glued^ past 10 μm away from the posterior. This plot does not effectively make the argument that arp1^4D2^ disrupts delivery of p150^Glued^ to the posterior, since no direct comparison of intensity is provided between WT and mutant. It appears that the data may be normalized to the maximum value in each case, but this just demonstrates that there is a more diffuse signal, rather than that the integrated intensity of p150^Glued^ is higher in WT as compared to mutant in the zone from 0-15 μm from the posterior. In the reviewer response, the authors state that "Even though p150^Glued^ /Dynactin is still present in the "anti catastrophe" area in the mutant, it cannot stabilise microtubule plus ends because of the mutation in Arp1." This statement seems to be at odds with the title of Figure 3, which reads "arp1^4D2^ disrupts the delivery of p150^Glued^, the dynein heavy chain and kinesin-βgal to the posterior." More importantly, this statement seems to propose an entirely different model than the primary model that is put forward in the paper.

This comment would be valid if our model were that the *arp1*^4D2^ mutant causes its phenotype by disrupting the delivery of Dynactin to the posterior, but this is not the case. We propose that *arp1*^4D2^ disrupts the ability of Dynactin to act as an anti-catastrophe factor at growing microtubule plus ends, which results in a loss of the long microtubules that extend all of the way to the posterior pole. The reduced posterior localisation of Dynactin is a consequence of the microtubule phenotype and not vice versa, as it is also a kinesin cargo that is transported along these microtubules. In support of this, we show that two other kinesin cargoes that are independent of Dynactin behave the same way (*oskar* mRNA and the kinesin-betagalactosidase fusion protein). Indeed, this transport of Dynactin by kinesin provides the basis for our proposal that Dynactin localisation provides positive feedback to the system, as its posterior enrichment will increase its anti-catastrophe effects in this region, allowing the microtubules to grow longer and localise more Dynactin.

The purpose of Figure 3 is to show that the localisation of several kinesin cargoes is abnormal in the *arp1*^4D2^ mutant. Figure 3 provides the quantification of these phenotypes and the figure legend provides the statistical analyses. We believe that the data provided in Figure 3 are in full support of our proposed model. We agree that the word "disrupt” in the Figure 3 title might be misleading. We have changed it to “affects”.

One way of addressing this concern is to quantify the intensity of the signal at the posterior in the wild-type and mutant from several samples per genotype with the same acquisition settings. If the quantification of the data would not support the proposed mechanism, the model might require significant adjustments.

We have not argued that there is less Dynactin in the posterior region in the *arp1*^4D2^ mutant, but merely show that it is more diffuse, because it is not being delivered to the posterior pole. Since none of our conclusions depend on the quantification of the amount of Dynactin posteriorly, we have not performed this additional analysis.

2) Figure 7 would be much stronger if the authors were able to also measure the lifetime of EB1 comets outside of the posterior region – their model is that the catastrophe rate is specifically altered near to the posterior, and so it seems that the lifetimes in Figure 7 should be measured both near to the posterior and away from the posterior – if the effect of the mutant is only at the posterior, then the difference in lifetimes should disappear for measurements away from the posterior.

We agree that the comparison of EB1 comet lifetimes at the posterior and more anteriorly would make 7E much stronger. However, we could not find a robust method to analyse microtubule lifetimes in more anterior regions of the oocyte. Firstly, the oocyte is much thicker in the middle compared to the posterior region. This reduces the imaging quality and dramatically increases the chance that a comet moves out of the imaging plane. Secondly, because the ncMTOCs are restricted to the anterior/lateral cortex, this region contains a much higher density of microtubules, most of which are growing at an angle to the anterior/posterior imaging plane and therefore visible for only part of their lifespan. By contrast, the posterior region lacks cortical ncMTOCs and the only microtubules to grow into this region must already be oriented close to the anterior/posterior imaging plane. This means that they disappear by going out focus at a much lower frequency, allowing us to collect meaningful data on their actual lifetimes.